# High-resolution maps show that rubber causes substantial deforestation

Yunxia Wang[1✉], Peter M. Hollingsworth[1], Deli Zhai[2], Christopher D. West[3], Jonathan M. H. Green[3], Huafang Chen[4,5], Kaspar Hurni[6,7], Yufang Su[8,11], Eleanor Warren-Thomas[9,10], Jianchu Xu[4,11] & Antje Ahrends[1✉]

Understanding the effects of cash crop expansion on natural forest is of fundamental importance. However, for most crops there are no remotely sensed global maps[1], and global deforestation impacts are estimated using models and extrapolations. Natural rubber is an example of a principal commodity for which deforestation impacts have been highly uncertain, with estimates differing more than fivefold[1–4]. Here we harnessed Earth observation satellite data and cloud computing[5] to produce high-resolution maps of rubber (10 m pixel size) and associated deforestation (30 m pixel size) for Southeast Asia. Our maps indicate that rubber-related forest loss has been substantially underestimated in policy, by the public and in recent reports[6–8]. Our direct remotely sensed observations show that deforestation for rubber is at least twofold to threefold higher than suggested by figures now widely used for setting policy[4]. With more than 4 million hectares of forest loss for rubber since 1993 (at least 2 million hectares since 2000) and more than 1 million hectares of rubber plantations established in Key Biodiversity Areas, the effects of rubber on biodiversity and ecosystem services in Southeast Asia could be extensive. Thus, rubber deserves more attention in domestic policy, within trade agreements and in incoming due-diligence legislation.

Around 90–99% of tropical deforestation is linked to the production of global commodities such as beef, soy, oil palm, natural rubber, coffee and cocoa[9]. Understanding the effects of individual commodities on natural forests is of fundamental importance for targeted policies and interventions. However, with relatively few exceptions—most notably oil palm and soy[1,10]—directly observed global or regional maps derived from satellite imagery are unavailable for most commodities. Instead, commodity-specific global deforestation is typically estimated using models[11,12] and extrapolations[13,14] with large levels of uncertainty.

Natural rubber is an example of a commodity whose effects on forests have remained poorly understood despite its economic importance[15] and the potential for widespread deforestation, land degradation and biodiversity loss[13,16–21]. Natural rubber is used in the manufacture of at least 1 billion tyres per year[15,22], and continued and increasing global demand is driving land use conversion in producer countries[14]. Production is primarily located in Southeast Asia (over 90% of the global production[23]), with the remainder coming from South and Central America and more recently also West and Central Africa[24]. Rubber is produced from the latex of a tropical tree (*Hevea brasiliensis*) and the spectral signature of rubber plantations is similar to that of forest[25], making it challenging to identify conversion of natural forest to rubber plantations from space. In addition, around 85% of global natural rubber is produced by smallholders[26], meaning that the plantations are scattered and often below 5 ha in size, increasing the challenge of detecting them from satellite imagery or capturing them in other forms in national crop statistics. Consequently, the locations and impacts of rubber plantations are surrounded by uncertainty and estimates of rubber-driven deforestation differ by more than fivefold: from less than 1 million ha almost globally between 2005 and 2018[3] to more than 5 million ha between 2003 and 2014 in continental Southeast Asia alone[2]. Direct observations based on remote sensing have previously existed only for subsets of Southeast Asia[2,27,28], individual countries[1,29] or subnational areas[30], and most are outdated so do not reflect the current risk.

At present, the most widely used dataset to estimate global rubber-related deforestation has been derived using a 'land balance' model[11]. This model combines remotely sensed data on tree cover loss with non-spatial estimates of crop expansion, derived mainly from national-scale statistics. The 'land balance' approach means that tree cover loss is not spatially linked to commodity expansion and therefore is not a substitute for more accurate products that provide spatially explicit estimates of crop expansion into forest areas, as explicitly acknowledged by the authors[31]. The land balance-derived data[3,4] suggest that rubber is a relatively minor problem when compared to the impact of other main forest risk commodities, with soy and palm oil

[1]Royal Botanic Garden Edinburgh, Edinburgh, UK. [2]Key Laboratory of Tropical Forest Ecology, Xishuangbanna Tropical Botanical Garden, Chinese Academy of Sciences, Xishuangbanna, China. [3]Stockholm Environment Institute York, Department of Environment and Geography, University of York, York, UK. [4]Centre for Mountain Futures, Kunming Institute of Botany, Chinese Academy of Sciences, Kunming, China. [5]China Country Program, CIFOR-ICRAF, Kunming, China. [6]Centre for Development and Environment, University of Bern, Bern, Switzerland. [7]East-West Center, Honolulu, HI, USA. [8]Institute of Economics, Yunnan Academy of Social Sciences, Kunming, China. [9]School of Natural Sciences, College of Environmental Sciences and Engineering, Bangor University, Bangor, UK. [10]International Institute for Applied Systems Analysis (IIASA), Laxenburg, Austria. [11]Present address: China Country Program, CIFOR-ICRAF, Kunming, China. ✉e-mail: wangyx.tina@outlook.com; aahrends@rbge.ac.uk

**Table 1 | Area estimates of rubber plantations for individual countries in Southeast Asia**

| Country | Rubber (ha) | Rubber (%) | Rubber in KBA (ha) | Rubber (%) in KBA | FAO 2020 harvested rubber (ha)[23] | Rubber in 2018 (ha)[28] | Rubber in 2014 (ha)[2] | |
|---|---|---|---|---|---|---|---|---|
| Indonesia | 4,745,921 | 34% | 362,951 | 8% | 3,668,735 | NA | NA | NA |
| Thailand | 3,744,139 | 26% | 291,600 | 8% | 3,292,671 | 4,650,000 | 1,429,487 | 2,861,400* |
| Vietnam | 1,606,594 | 11% | 59,401 | 4% | 728,764 | 740,000 | 912,696 | 1,916,600* |
| China | 1,097,213 | 8% | 58,073 | 5% | 745,000 | NA | NA | NA |
| Malaysia | 985,335 | 7% | 49,391 | 5% | 1,106,861 | NA | NA | NA |
| Myanmar | 779,717 | 6% | 84,577 | 11% | 323,956 | 680,000 | NA | NA |
| Cambodia | 618,135 | 4% | 117,682 | 19% | 310,877 | 200,000 | 917,446 | 2,974,300* |
| Laos | 574,035 | 4% | 49,125 | 9% | NA | 700,000 | 260,471 | 765,600* |
| **Southeast Asia** | **14,151,090** | | **1,072,800** | **8%** | **10,176,864** | | | |
| | 24,587,796* ± 4,615,324 (95% CI) | | | | | | | |

For China, only the main production areas are included (Xishuangbanna and Hainan). Here, we present our most conservative figures (mapped area). The sample-based area estimate and its CI (following ref. 33; Supplementary Table 1) suggest that the rubber area may be higher (indicated by an asterisk). Reference 2 also derived standard mapped figures and sample-based area estimates (indicated by an asterisk). For Thailand, their figures only include northeast Thailand, and for Vietnam, only areas south of Hanoi. NA, not available.

accounting for seven and eight times more deforestation than rubber, respectively; and in UK imports[6] for 57 and 20 times more deforestation. This has contributed to the reduced attention that rubber has received as a driver of deforestation compared to other commodities and has led to extensive debate about the need to include rubber in policy, such as the European Union (EU) Deforestation Regulation[7] and secondary legislation associated with the UK Environment Act Schedule 17. However, given the inherent uncertainty in model-based estimates, there is an urgent need for robust evidence to provide guidance for policy interventions to avoid rubber being prematurely excluded from key policy processes and interventions.

Furthermore, monitoring the effectiveness of policy and compliance with legal and voluntary zero-deforestation commitments will need spatially explicit commodity production data. This is now highly relevant because, following prolonged uncertainty about the inclusion of rubber in the EU Deforestation Regulation, a recent trialogue (December 2022) reached agreement to extend the scope of the regulation to also include rubber (a preprint version of this manuscript (https://doi.org/10.1101/2022.12.03.518959) formed part of the evidence contributing to the trialogue), a decision adopted by the European Parliament on 19 April 2023. The ability to monitor rubber-related deforestation will be critical for the implementation of this legislation, for similar legislation potentially following in the United Kingdom and USA (for which relevant acts are now restricted to illegal deforestation) and for monitoring various private sector voluntary commitments such as those made under the auspices of the Global Platform for Sustainable Natural Rubber (GPSNR).

Here we present up-to-date analyses and provide Southeast Asia-wide maps of rubber and associated deforestation, encompassing more than 90% of natural rubber production volume. This is now possible thanks to increases in the resolution of Earth observation data, which can also capture smallholder plantations. We used the latest high-resolution Sentinel-2 imagery (at a spatial resolution of 10 m) to map the extent of rubber across all Southeast Asia in 2021. Our approach is based on the distinctive phenological signature of rubber plantations, which allows them to be distinguished from both evergreen (Extended Data Fig. 1) and deciduous (Extended Data Fig. 2) tropical forests on the basis of leaf fall and regrowth, which (particularly in mainland Southeast Asia) occur in specific time windows. To tackle the challenge of heavy cloud cover in the region we used multiyear imagery composites. For all areas identified as rubber in 2021 we assessed whether (and when) prior deforestation had occurred using historical Landsat imagery and a spectral-temporal segmentation algorithm (LandTrendr)[32]. The Landsat archive allowed us to track deforestation back to the early

1990s. We count only the first occurrence of deforestation to minimize the inclusion of plantation rotation. Here, we use the term 'deforestation', but it is of note that we track any type of tree cover loss since 1993. Thus, the rubber-related 'forest' loss quantified here can include the conversion or rotation of agroforests, plantation forests, agricultural tree crops and rubber itself if established in the 1980s and hence mature by 1993 (Supplementary Note). A graphical overview of our methods is available in Extended Data Fig. 3.

## Rubber map for Southeast Asia

According to our maps, mature rubber plantations occupied an area of 14.2 million hectares in Southeast Asia in 2021, with more than 70% of the production area situated in Indonesia, Thailand and Vietnam. Other notable areas were situated in China, Malaysia, Myanmar, Cambodia and Laos (Table 1 and Fig. 1a). This figure is conservative in that estimates based on reference ground data[33] suggest that rubber may occupy a larger area in Southeast Asia (Table 1 and Supplementary Table 1). The rubber maps achieved a good overall classification accuracy (OA = 0.95 ± 0.02 95% confidence interval (CI); Supplementary Table 1) with good accuracy and precision of estimates for mainland Southeast Asia (OA > 0.99 ± 0.01 95% CI; Supplementary Table 2) but higher omission errors and less overall accuracy for insular Southeast Asia (OA = 0.85 ± 0.06 95% CI; Supplementary Table 3). Here, limited seasonality (Extended Data Fig. 4) and greater heterogeneity in climatic conditions (Extended Data Fig. 5) mean that rubber phenology is less predictable, with trees defoliating at different times, exhibiting partial defoliation or no defoliation at all[34]. Hence, despite running the rubber detection algorithm separately for two different subregions to address the spatial heterogeneity in climate conditions (Extended Data Fig. 6), omission errors remain in insular Southeast Asia (Extended Data Figs. 7 and 8; see Methods). Overall, user's accuracy (the complement of commission error) was 0.99, and producer's accuracy (the complement of omission error) was 0.95 but dropped to 0.57 when based on estimated area. (When based on estimated area the error matrix and hence producer's accuracy are adjusted by area weights, calculated as the proportionate area occupied by the class[33], meaning that the complement of producer's accuracy measures potentially omitted area proportions.) The low producer's accuracy when based on estimated area is in part due to us erring on the side of omission errors (mainly affecting insular Southeast Asia) and also because rubber occupies a proportionately small area compared to the class it is separated from (all other tree cover), meaning that any rubber point erroneously mapped as other tree cover had a large influence on

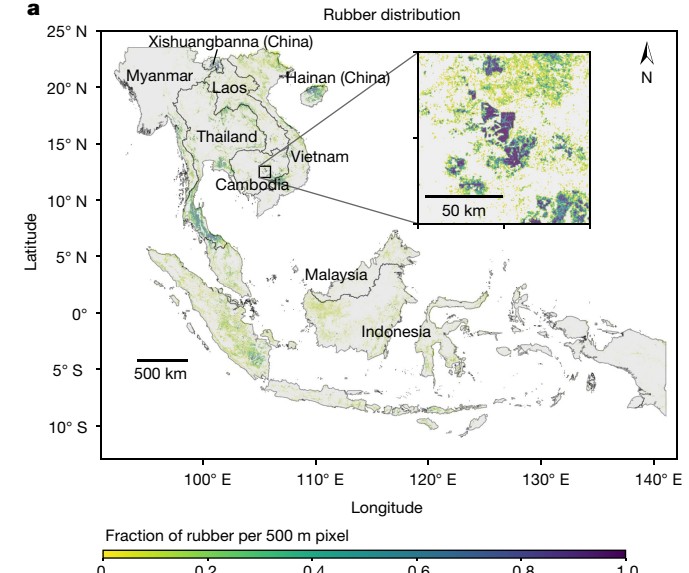

**Fig. 1 | Rubber distribution in 2021 and associated deforestation across Southeast Asia. a,b,** Rubber distribution (**a**) and associated deforestation (**b**). For better visualization, the rubber map (**a**) was aggregated to 500 m pixel size by calculating the proportion of 10 m rubber pixels in each 500 m pixel and the rubber-related deforestation map (**b**) was aggregated to 500 m pixel size by calculating the proportion of 30 m deforestation pixels within each 500 m pixel. The maps in their original resolution are available at https://wangyxtina.

users.earthengine.app/view/rubberdeforestationfig1. The area mapped as rubber is conservative and has higher accuracy for mainland Southeast Asia than for insular Southeast Asia (here defined as all of Malaysia and Indonesia), for which omission errors were higher (Supplementary Tables 1–3). Source of administrative boundaries: the Global Administrative Unit Layers (GAUL) dataset, implemented by FAO within the CountrySTAT and Agricultural Market Information System projects.

estimated rubber area (Supplementary Table 1). Although we present both mapped and estimated area (Table 1), we emphasize the more conservative (mapped) estimate.

Our mapped estimate of 14.2 million ha rubber in Southeast Asia is consistent with the sum of national statistics reported to the Forest and Agriculture Organization of the United Nations (FAO), according to which the total area of harvested rubber in Indonesia, Thailand, Vietnam, China, Malaysia, Myanmar, Cambodia and Laos was 10.18 million ha in 2020[23]. Owing to the now low global rubber price many plantations may not be harvested, meaning that, although our mean estimate is higher than the values reported to the FAO, there is a broad alignment. Our estimates are also generally within the bounds estimated by two other recent remote sensing studies for rubber[2,28] (Table 1).

## Substantial deforestation due to rubber

We used time-series Landsat imagery to identify the deforestation date for all areas classed as rubber in 2021 in two categories: 1993–2000 and 2001–2016 (overall classification accuracy of 0.85 ± 0.09 95% CI; Supplementary Table 4). For this we used the LandTrendr algorithm[35], which identifies breakpoints in the pixels' spectral history. Here, we tracked the largest breakpoint in the normalized burn ratio (NBR), indicative of a sudden change from forest or other types of tree cover to bare and/or burnt ground (Extended Data Fig. 9). We used only the first main breakpoint, going as far back in time as the imagery allows (early 1990s), meaning that we include rotational plantation clearance into the deforestation estimate only if these plantations were established in the 1980s and hence detectable as mature tree cover by the early 1990s. In addition, we count pixels as deforested only if their previous NBR was above a threshold of 0.6 to minimize the inclusion of pixels that may have been deforested or degraded before the 1990s.

Our data show that rubber led to substantial deforestation across all of Southeast Asia (Fig. 1b). In total, we estimate that 4.1 million ha of forest were cleared for rubber between 1993 and 2016. This is a conservative estimate for two reasons: (1) we map deforestation only for the

area mapped as rubber in 2021, meaning that if our rubber area map is conservative (see above), so is our map of rubber-related deforestation and (2) the NBR threshold we use may lead to underestimated deforestation in areas with naturally drier vegetation, more bare ground and/or regular fires. Removing the threshold leads to an estimate of almost 6 million ha of forest loss.

According to our maps, almost three-quarters of this forest clearance occurred since 2001 (3 million ha). Sample-based area estimates (Supplementary Table 4) suggest that the deforested area since 2000 may have been somewhat lower (2.5 million ha ± 0.35 million ha 95% CI), but overall our results suggest that rubber-related deforestation is not just a historic problem and that substantial deforestation occurred after 2000. In addition, more than 1 million ha of rubber plantations in 2021 were situated in Key Biodiversity Areas (KBAs)[36,37], which are globally important for the conservation of biodiversity (Table 1).

In terms of individual countries, both historically and since 2001, deforestation was highest in Indonesia, followed by Thailand and Malaysia (Figs. 2 and 3). Although these three countries accounted for more than two-thirds of total rubber-related deforestation in Southeast Asia during 2001–2016, substantial deforestation also occurred in Cambodia since 2001, where more than 40% of rubber plantations were associated with deforestation (Fig. 2) and 19% of rubber area was situated in KBAs (Table 1).

## Rubber deforestation is underestimated

Recent estimates of deforestation embedded in rubber, intended to inform policy in the EU[7], G7 (ref. 8) and the United Kingdom[6], all used the data generated by ref. 11, which place total rubber-related deforestation between 2005 and 2017 at below 700,000 ha (in 135 countries, including all principal rubber producers, except China and Laos). Translating to an average annual deforestation of 53,000 ha (Table 2), these estimates lie several-fold below the estimates of this and other studies on the basis of spatially explicit data—in the case of Cambodia, several hundredfold (Table 2). A revision of the data from ref. 3 now provides an almost

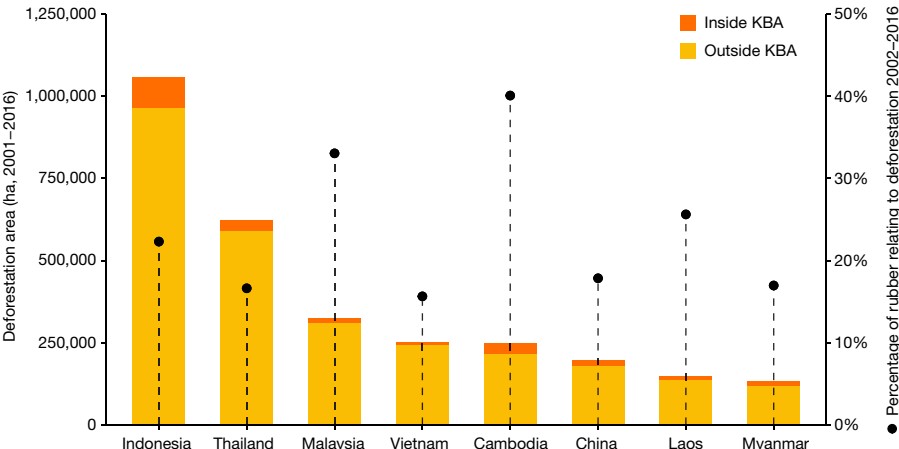

**Fig. 2 | Area of rubber-related deforestation between 2001 and 2016 for individual countries in Southeast Asia.** The bars show the cumulative area of deforestation (2001–2016) for rubber plantations in 2021. Orange areas are the fraction of deforestation that occurred inside KBAs[35]. The circles show the percentage of the total national rubber area in 2021 that was associated with deforestation between 2001 and 2016 (the percentage is given on the second y axis). The figures for China include only its main production areas (Xishuangbanna and Hainan).

30-fold higher estimate for deforestation in Cambodia (Table 2) but still places total quasiglobal rubber-related deforestation between 2005 and 2018 below 1 million ha. By contrast, the World Resources Institute[1] estimated that rubber replaced 2.1 million ha of forest during 2001–2015 in just seven countries, which account for less than half of global natural rubber production, and ref. 2 estimated that rubber replaced more than 5 million ha of forest in continental Southeast Asia alone. Although our estimates are conservative compared to these other estimates and because none of the figures can be directly compared as they refer to somewhat different time periods and different definitions of forest, it is of critical note that even our lower 95% CI still greatly exceeds (more than double) the model-based estimates now widely used to guide policy and to calculate deforestation footprints. Furthermore, even if we replaced our estimates for Indonesia and Malaysia with those of ref. 11, the two countries in which ref. 11 attempted to exclude plantation rotation from deforestation totals,

our annual rubber-deforestation totals would still be more than twice as high (Supplementary Note).

## Discussion

Here we provide high-resolution maps for rubber and associated deforestation between 1993 and 2016 for all Southeast Asia. We show that rubber has led to several million hectares of deforestation and that the global data[3,4] now widely used in setting deforestation policies are likely to severely underestimate the scale of the problem. Although very helpful for providing a holistic assessment of the role of agricultural commodities in driving tropical and subtropical deforestation across the globe, these previous and other model-based data are not a substitute for spatially explicit estimates of crop expansion into natural forests[31]. Our estimates lie several-fold above these data despite covering only Southeast Asia and not, for example, West and Central Africa,

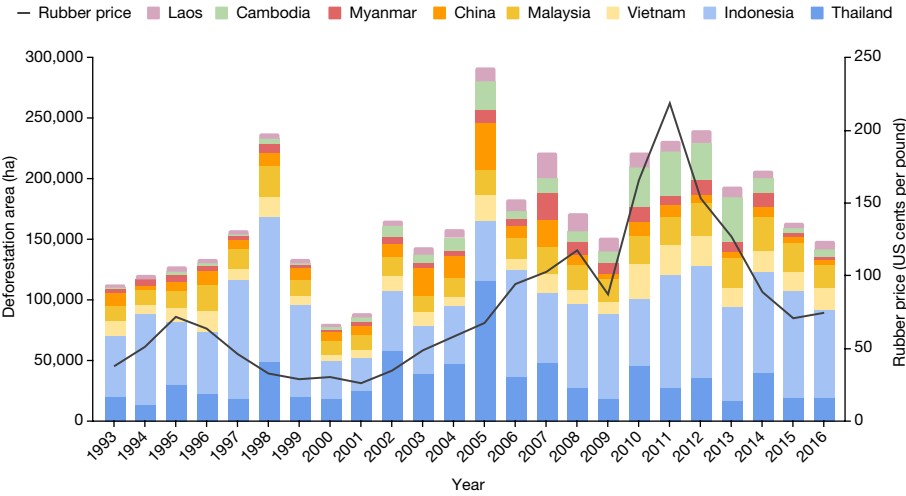

**Fig. 3 | Total area of rubber-related deforestation in Southeast Asia between 1993 and 2016.** The colours show the fraction of overall deforestation that occurred in individual countries. Although most deforestation occurred in Indonesia and Thailand and the deforestation trends are similar across countries, the fraction of deforestation occurring in mainland Southeast Asia (mainly Cambodia) has increased over the past decade. The rates of rubber expansion and associated deforestation involve decisions taken by millions of

actors and are influenced by complex and interlinked drivers such as national policies and subsidies, prices for other crops and the availability of extension services and infrastructure. However, it is noteworthy that in some countries (for example, Cambodia[29] and Vietnam) rates of rubber-related deforestation increased alongside global rubber price increases after 2000 (black line, second y axis; source: International Monetary Fund, accessed at https://fred.stlouisfed.org/series/PRUBBUSDM).

**Table 2 | Comparison of rubber-related deforestation estimates generated by this and other studies**

| | Method | Definition of 'forest' | Time period | Reference area | Rubber-related deforestation in 1,000 ha yr$^{-1}$ | | | | |
|---|---|---|---|---|---|---|---|---|---|
| | | | | | Total in reference area | Indonesia | Thailand | Malaysia | Cambodia |
| **Ref. 4** | **Land balance model** | **Tree cover greater than or equal to 25% (ref. 51)** | **2005–2017** | **135 tropical countries, including all chief rubber producers (except China and Laos)** | **53** | **22** | **9** | **5** | **0.1** |
| Ref. 3 | | | 2005–2018 | | 52 | 23 | 6 | 5 | 3 |
| Ref. 1 | Mix of spatially explicit data | Tree cover greater than or equal to 30% (ref. 51) | 2001–2015 | Brazil, Cambodia, Cameroon, Democratic Republic of the Congo, India, Indonesia and Malaysia | 140 | 64 | NA | 48 | 22 |
| Ref. 2 | Remote sensing | Internal classifier | 2003–2014 | Mainland Southeast Asia | 135 | NA | NA | NA | 69 |
| | | | | | 437* | | | | 232* |
| Ref. 29 | Remote sensing | Tree cover greater than or equal to 10% (ref. 52) | 2001–2015 | Cambodia | | NA | NA | NA | 34 |
| This study | Remote sensing | ESA WorldCover 10 m 2020 v.100 (tree cover greater than or equal to 10%) | 2001–2016 (baseline 1993) | Southeast Asia | 186 | 66 | 39 | 20 | 15 |
| | | | | | 156*±22 | NA | NA | NA | NA |

The dataset in bold (first row) has been used to guide deforestation policy[7] and to calculate the imported deforestation of individual countries[6,8]. In this study, we use a conservative baseline of 1993. The earliest baseline in other studies is 2000 and hence other studies will include more plantation rotation. The different base lines also mean that our estimates cannot easily be set into the context of overall deforestation in Southeast Asia (estimated to be 3.22 million ha yr$^{-1}$ between 2001 and 2019[18]). At face value our rubber deforestation estimates account for 5–6% of that figure but this is very conservative as the overall figure is derived using a baseline of 2000 and hence includes more plantation rotation (of rubber and other types of tree cover). Sample-based area estimates for this study (following ref. 33) and for ref. 2 are indicated by an asterisk.

where there has been substantial recent rubber expansion, probably driving deforestation[24].

Owing to the heterogenous data landscape with greatly variable accuracy across crops, the effects of crops on deforestation cannot be reliably compared. The findings of this study would place rubber deforestation above the effects found for coffee and, contrary to what has been previously assumed, above the effects of cocoa[1,4]. The rubber impact is still lower than the impact of oil palm, but not by a factor of 8–10 as has been previously suggested[1,4] and instead only by a factor of 2.5–4.0 (also noting that here we are comparing our data for Southeast Asia only with global estimates for these other crops). However, these comparisons are difficult to make, not least because the estimated impacts of cocoa also differ threefold between studies[1,4], with cocoa being another example of a crop for which there are no global remotely sensed maps.

Our map of rubber extent is likely to be conservative. First, we used 2021 as the reference year and hence do not capture deforestation for rubber if, by 2021, the rubber plantation had been converted to a different land use. Because there was a rubber price boom in the first decade of this millennium, followed by a price crash since 2011[38], it is possible that in the meantime some rubber area has been converted to other, more lucrative, land uses[38], which will not be included in our estimates. Second, ground reference data indicate that we err on the side of omission errors, with sample-based area estimates[33] suggesting that the rubber area could be substantially larger (Supplementary Table 1), particularly in insular Southeast Asia. This is because the limited seasonality of the equatorial climate precludes a strong and predictable phenological response of rubber in insular Southeast Asia[34]. Furthermore, insular Southeast Asia has more persistent cloud cover than mainland Southeast Asia, with 7% and 10% of the study area in Indonesia and Malaysia, respectively, lacking clear Sentinel-2 images (Supplementary Table 7). Consequently, our maps are more accurate for mainland Southeast Asia than for insular Southeast Asia (Supplementary Tables 2 and 3), where rubber area (and hence associated deforestation) may be underestimated. Any comparisons by country or other spatial units across these two subregions thus need to be done with caution in the light of this limitation. Third, we used the European Space Agency (ESA) global tree cover map[39] as a mask for mapping rubber plantations. If rubber areas were not picked up as tree cover by this map, they are also excluded from our estimates. Finally, we map only mature rubber; younger rubber plantations (around less than 5 years old) are excluded. Our algorithm is also unlikely to detect diseased rubber if this is manifested as unseasonal leaf shedding, or rubber-based agroforestry systems and 'jungle' rubber[40] (now economically marginal[41]) unless rubber is the dominant component of the canopy. If our rubber map is conservative, mapped deforestation will also be conservative, as deforestation detection was restricted to areas mapped as rubber.

We have considered and accommodated possible areas of ambiguity that might otherwise lead to an overestimation of deforestation using our method. First, rotational plantation and tree crop clearing and replanting may erroneously be classed as deforestation. This is a key issue, which is notoriously difficult to address and hence also affects other studies[1,11] (Supplementary Note). The issue is likely to be particularly important in Indonesia, Malaysia and Thailand, where rubber and other plantations have a longer history of planting. To address this, we use the first deforestation date and ignore subsequent pixel changes, meaning that this problem would apply only to plantations and tree crops established before, and mature by, 1993. This baseline is relatively conservative. In addition, we set a strict NBR threshold (indicative of 'green and healthy' vegetation) that pixels had to exceed before counting as deforested; relaxing that threshold leads to substantially higher deforestation estimates. Second, deforestation may have occurred for a different land use, with the area subsequently converted to rubber. This may particularly be the case in more marginal climates in mainland Southeast Asia where rubber expansion is more recent[16] (for example, deforestation in northern Vietnam in the 1990s may have mainly occurred for industrial forestry, with rubber replacing forestry plantations more recently). However, the issue will be smaller for rubber than for plantations such as oil palm, which boomed and expanded more recently[42], possibly replacing other land uses in addition to forests. Rubber is a crop with a longer history in the area and a greater plantation longevity of around 25 years[30]. Third, the vegetation in some pixels may have undergone some type of disturbance in the rubber defoliation time window, followed by regrowth in the rubber refoliation window, leading to them having the

characteristic phenology signature of rubber and erroneously being classed as such. To exclude such pixels and increase the accuracy of our analysis we created a 'disturbance' mask (Methods). Thus overall, we consider our estimates of deforestation due to rubber plantations more likely to be an underestimate than an overestimate of the scale of the issue.

The current estimates for deforestation caused by rubber[3,4] used for policy considerations in the EU[7] and the United Kingdom[6] are based on a land balance model[11,12]. Such models typically allocate total deforestation area to different commodities on the basis of national (or subnational, for example in the case of this model for Brazil and Indonesia) reports of crop expansion[11]. This can lead to substantial overestimates or underestimates of the role of different crops in driving deforestation[31]. First, crop expansion statistics are hampered by uncertainties and inconsistent reporting across crops and countries. Second, although the total area of a crop can remain stable, its actual place of occupancy may change[31]. This is highly relevant to rubber as oil palm has expanded into traditional rubber growing areas[43,44], with new compensatory rubber plantations being established elsewhere, for example, in uplands[18,30] and often climatically marginal areas[16], where they may be associated with deforestation. In fact, the land balance model[4] includes a large amount of unattributed deforestation that could not be explained by crop expansion statistics. Our higher rubber deforestation estimates could help to explain some of this unattributed deforestation. In summary, while the use of extrapolation[13,14] and model-based[11,12] approaches provides some form of estimation for the extent of deforestation due to rubber plantations, we advocate caution in its interpretation. Instead, where available, we argue for the use of results from direct observations of the dynamics of crop production systems (for example, using remotely sensed satellite imagery), thereby greatly increasing the accuracy of deforestation estimates.

In terms of future projections of the impact of rubber and the time-critical need for deforestation legislation, it is likely that demand for natural rubber will continue to increase[15]. Synthetic alternatives or other natural sources are not a perfect substitute[45,46] and, being based on petrochemicals primarily derived from crude oil, they are also considered more environmentally harmful. Natural rubber, on the other hand, is a renewable resource with the potential to contribute to climate change mitigation[47] and benefit the livelihoods of smallholder farmers[48]. However, if not regulated carefully, rubber growing can have severe negative consequences for livelihoods[26,49] and lead to environmental degradation[13,16–21] and biodiversity loss[41]. These impacts are often concealed to consumers, with natural rubber products being marketed as 'sustainable products made from trees'. Our deforestation data also suggest that the assumed 'breathing space'[38] generated by the now low rubber price may be false, with continued (and volatile) deforestation for rubber since 2011, a problem that could increase if rubber prices rise again.

Given the substantial rubber-related deforestation demonstrated here, it is encouraging that rubber is beginning to be included in relevant policy debates, with the last-minute inclusion of rubber in the scope of the EU Deforestation Regulation. Initiatives such as the GPSNR, a multistakeholder membership organization committed to transparent improvements in socioeconomic and environmental performance of the natural rubber value chain, are also requiring members to address deforestation. A frequently voiced concern is that rubber supply chains are difficult to trace and that deforestation regulations place a disproportionate burden on rubber operators. Contrary to oil palm, for which there is a limited time window (about 24 h) between harvest and processing at mills, unprocessed rubber has greater longevity, allowing transport over several hundred kilometres and exchange between several aggregators before arrival at processing facilities[50], presenting traceability challenges. Another critically important point is the need to ensure that smallholders are not disadvantaged by deforestation regulations, as, contrary to larger companies, they may not be able to afford the premiums for certified sustainable production. Although concerns about the potential marginalization of smallholders apply to all commodities, it is a particularly important consideration for commodities that are strongly linked to smallholder livelihoods and development prospects, such as rubber. Recent initiatives, for example by the Forest Stewardship Council, have demonstrated that the challenges can be overcome when farmers are organized in groups, with an extra benefit being that farmer cooperatives can negotiate a joint price to buffer their livelihoods against the volatile global rubber price. In addition, whilst supply chains are indeed complex and challenging to trace, the high-end rubber processing side is dominated by very few and identifiable actors. Around 70% of the global natural rubber production is used in tyres with a few main companies accounting for most consumption[15], many of which are already part of the GPSNR.

Further work is needed to make connections between rubber-driven deforestation and specific supply chains but, in the absence of such information, it should be assumed that main importers of rubber such as the EU are substantially exposed to rubber-related deforestation. In addition, the lack of traceability information at present provides a further argument for the inclusion of rubber in regulatory processes to drive traceability efforts and to provide an opportunity for supply chains to support sustainable production.

In summary, we believe that rubber merits more consideration in policies and processes that aim to reduce commodity-driven deforestation and that it is vitally important to use the best available evidence on the scale of the problem. The issue outlined here for rubber is of fundamental importance in its own right because rubber is responsible for millions of hectares of deforestation. However, we also highlight the wider need to enhance the evidence base available to inform policy decisions and to aid their implementation. There is an opportunity for increased clarity and rigorous quantification of the extent of environmental degradation caused by main cash crops that is increasingly possible using remotely sensed Earth observation.

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

## Methods

Here, we used Sentinel-2 imagery to produce a map of rubber plantations for all Southeast Asia in 2021, and we mapped the occurrence and the timing of deforestation for these plantations on the basis of time-series data from Landsat images (1993–2016). An overview of the Methods is presented in Extended Data Fig. 3.

### Sentinel-2 imagery

Sentinel-2 is an optical multispectral imaging mission from the Copernicus Programme headed by the European Commission in partnership with ESA[53]. It acquires very high-resolution multispectral imagery with a global revisit frequency of 5 days. In this study, we used the Sentinel-2 level-2A Surface Reflectance imagery for 2020–2022 obtained through Google Earth Engine[5] to map the extent of rubber plantations in Southeast Asia in 2021. Sentinel-2 Surface Reflectance imagery has been corrected for atmospheric influences with the Sen2Cor processor algorithm[54–56]. To remove clouds and cloud shadows, we used the QA60 cloud mask band and Sentinel-2 cloud probability datasets[55] in which pixels with cloud probability greater than 50% are considered as clouds. Cloud shadows are defined as areas of cloud projection intersection with low-reflectance, near-infrared pixels. Full details are available at https://developers.google.com/earth-engine/tutorials/community/sentinel-2-s2cloudless. Cloud cover was a small issue in mainland Southeast Asia but presented greater challenges in insular Southeast Asia, affecting 7% and 10% of the study area in Indonesia and Malaysia, respectively (Supplementary Table 7).

For each image, we selected ten bands and computed seven spectral indices. The bands comprised four 10 m resolution bands (blue, B2; green, B3; red, B4; and near-infrared, B8) and six 20 m resolution bands (red-edge bands[57], B5, B6, B7 and B8A; short-wave infrared bands, B11 and B12). The seven spectral indices were normalized difference vegetation index (NDVI), normalized difference water index (NDWI), renormalization of vegetation moisture index (RVMI), NBR, modified NBR (MNBR), soil-adjusted vegetation index (SAVI) and enhanced vegetation index (EVI). All bands and spectral indices were resampled to 10 m resolution for further analysis. Working with a 10 m resolution instead of a 20 m resolution allowed us to take advantage of the high resolution of key bands (for example, the NDVI component bands B4 and B8) to capture smallholder plantations (often less than 1 ha in size) as best as possible.

The equations used for calculating the spectral indices are as follows:

$$NDVI = \frac{B8 - B4}{B8 + B4} \tag{1}$$

$$NDWI = \frac{B8 - B11}{B8 + B11} \tag{2}$$

$$RVMI = \frac{NDVI - NDWI}{NDVI + NDWI} \tag{3}$$

$$NBR = \frac{B8 - B12}{B8 + B12} \tag{4}$$

$$MNBR = \frac{B8 - (B11 + B12)}{B8 + B11 + B12} \tag{5}$$

$$SAVI = \frac{1.5 \times (B8 - B4)}{(B8 + B4 + 0.5)} \tag{6}$$

$$EVI = \frac{2.5 \times (B8 - B4)}{(B8 + 6 \times B4 - 7.5 \times B2 + 1)} \tag{7}$$

### Mapping the extent of rubber plantations

We designed a new phenology-based methodology to map rubber plantations across Southeast Asia. Unlike evergreen and deciduous tropical forest and most other tree plantations present in the region, rubber plantations shed their leaves during the dry season and subsequently regain their leaves before the onset of the wet season. Whether this is primarily a response to drought or cold stress is the subject of ongoing research[58,59] but, particularly in mainland Southeast Asia, the cold and dry seasons coincide, meaning that, here, the lack of mechanistic understanding of this phenological response does not preclude identifying the time window of its occurrence.

While mainland Southeast Asia is characterized by a seasonal monsoonal climate, insular Southeast Asia is less seasonal and the onset of a dry season, if present, mostly falls into a different time of year compared to mainland Southeast Asia (Extended Data Fig. 5). Therefore, we divided the region into two subregions (Extended Data Fig. 6). In mainland Southeast Asia, the northeast monsoon brings dry and cool continental air[60] and rubber defoliation generally occurs during January–February with subsequent refoliation during March–April (Extended Data Fig. 1). This distinct signature also allows the separation of rubber from deciduous forest, which is present in much of mainland Southeast Asia: leaf regrowth in other species in deciduous forest mainly coincides with the onset of the wet season in May (Extended Data Fig. 2).

In contrast to mainland Southeast Asia, large parts of insular Southeast Asia do receive rainfall during the northeast monsoon with the southwesterly flowing air masses gathering moisture as they pass over the warm sea. Instead, there can be a dry season during the southwest monsoon (May to September) when the air masses reverse and the northeasterly blowing winds bring dry air from the Australian continent[60]. However, in the equatorial maritime climate the dry season tends to be neither prolonged nor distinctive (Extended Data Fig. 4) and soil moisture can remain stable or at least above critical levels[34].

Originating from the Brazilian Amazon, the deciduous behaviour of *H. brasiliensis* is thought to have evolved as an adaptive strategy for drought or more generally stress avoidance[59]. Consequently, in years or areas where there is no clear-cut stress in the form of a distinctive dry and/or cold season, leaf shedding will only be partial, not take place at all and/or will be influenced by micrometeorological conditions with trees defoliating asynchronously even within the same stand[34]. Few reports exist on rubber phenology in insular Southeast Asia. The limited available evidence[34,61–65] (covering about 18 sites, which are spatially biased towards the main rubber growing areas Sumatra and Malay Peninsula, with only one report for Borneo and none for islands further east) suggests that, where there is a predictable defoliation window, it generally occurs during January–February (Malay Peninsula and northern Sumatra) or during June–September (further south).

Because the divergent defoliation patterns described in the available literature mainly affect Indonesia and as, owing to consistently high temperatures, stress, if present, is likely to occur in the form of drought, we delineated two climatic subzones as follows: we mapped average monthly precipitation[66] across Indonesia and identified the driest month for each pixel (around 1 × 1 km); we then delineated all pixels with the driest month between June and September as a separate subregion (region B, where defoliation was assumed to take place June–September with subsequent refoliation during October–December). The remaining pixels and all of Malaysia and mainland Southeast Asia were assigned to region A, where defoliation was assumed to take place between January and February with subsequent refoliation during March–April (Extended Data Fig. 6).

The lack of distinctive seasonality near the equator means that inaccuracy of our classification was greatest near the equator (Extended Data Figs. 7 and 8) and mainly manifested in omission errors (3% of our 661 ground reference points used for validation were false negatives

and only 0.3% were false positives; of the false negatives, 95% occurred in insular Southeast Asia). Beyond about 7° N the climate becomes more continental with clear-cut seasonality and no more false negatives were recorded.

The unique phenology of rubber, where exhibited, thus makes rubber distinguishable from other tree cover using satellite imagery. Here we used a tree cover mask from the ESA global land cover map[39] (the ESA WorldCover 10 m 2020 product) as a base map for classifying tree cover into rubber and other tree cover based on the spectral differences described above. According to an independent evaluation[67] the ESA global land cover map achieves reasonably good accuracies for tree cover (user's accuracy of 80.1 ± 0.1 95% CI and producer's accuracy of 89.9 ± 0.1 95% CI). For the defoliation stage, we generated a composite image using a 15% NDVI percentile threshold of all images acquired during January and February in 2021 and 2022 for region A and during June–September in 2020 and 2021 for region B. For the refoliation stage, we used the 85% NDVI percentile as a threshold to generate a composite of all images acquired during March and April in 2021 and 2022 for region A and during October–December in 2020 and 2021 for region B. This was to reduce noise generated by remaining clouds and shadows. Each composite image contained 17 variables, including 10 spectral bands and 7 spectral indices (see section on 'Sentinel-2 imagery' above).

The classification was produced using a random forest machine learning algorithm. For hyperparameter settings and a summary of individual variable contributions to the classification, see Supplementary Tables 8 and 9. We collected a total of 3,826 reference sample points (2,010 for rubber and 1,816 for evergreen forest; Extended Data Fig. 6) and randomly split them into 80% and 20% for training and testing the random forest classifier, respectively. This left us with about 700 points for testing; following the equations by ref. 33 we estimated that a sample size of $n = 441$ was sufficient for achieving a standard error of the overall accuracy of s.e. = 0.01. Of these more than 3,800 points, 2,000 were based on randomly sampled reference ground data collected by the World Agroforestry Centre in 2010, covering the entire region and consisting of a mix of field data and visually interpreted very high-resolution satellite data. We revised the classification for these points for 2021 following a visual interpretation protocol (see below). The remainder were points from randomly sampled reference ground data covering mainland Southeast Asia[68] and Xishuangbanna, China[69]. With more than 50% of the points used in this study collected in the field, their classification is likely to be very accurate. However, any field data will to some extent suffer from an accessibility bias with potential implications for accuracy and area estimation, which we further discuss below.

The visual interpretation process was carried out by two interpreters using Collect Earth Online[70–72] (CEO) and Google Earth Pro[73] (Supplementary Fig. 1). Google Earth Pro provided access to high and very high-resolution imagery with acquisition dates, and a custom-built project in CEO provided access to very high-resolution Mapbox Satellite imagery base maps, 2021 monthly Planet NICFI images (Norway's International Climate and Forests Initiative satellite data program) and yearly composite images for January–February and March–April from Sentinel-2 (2017–2021)[1] and Landsat-5-7-8 (1988–2016; courtesy of US Geological Survey). First, we assigned each sample point to a land cover class for the year 2021. Second, if the land cover was rubber, we identified the deforestation date for that point using historical Landsat images. Where available, more very high-resolution imagery from Google Earth was used to facilitate the interpretation process.

Disturbances such as degradation or plantation removal can potentially produce similar spectral features to rubber phenology, leading to commission errors. To reduce commission errors, we removed all rubber pixels where this may have occurred using a 2021 primary forest mask and a no-disturbance mask (Extended Data Fig. 3). The 2021

primary forest mask was created by using the 2001 primary forest layer from ref. 74 and removing areas of subsequent forest loss between 2000 and 2021 (Hansen Global Forest Change v.1.9)[51]. The no-disturbance mask was generated with the following steps: (1) calculate the NBR index (equation (4)) for all Sentinel-2 images between 2019 and 2021; (2) create 3-year NBR median composites for March–June, July–September and October–December (region A) or January–May and October–December (region B) (yielding three composites for region A and two composites for region B); (3) extract the values of NBR composites for all the rubber samples; (4) plot the NBR values and calculate the 5% percentile thresholds for individual composites, meaning 95% of NBR values of rubber samples are above these thresholds; and (5) apply the thresholds to all three (region A) or two (region B) NBR composite images, resulting in five binary images (1, no disturbance; 0, potential disturbance). If a pixel was classed as 1 in all three (region A) or two (region B) binary images, it was considered as not disturbed. A 5 × 5 pixel majority filter was applied to the no-disturbance mask to remove isolated pixels.

The accuracy of the final map was evaluated using the remaining 20% of the reference ground data points ($n = 661$), following standard good practices[33] (Supplementary Table 1). Sample-based area estimates[33] suggested that the rubber area could be substantially larger than mapped (Supplementary Table 1), particularly in insular Southeast Asia (Supplementary Table 3). This is likely to be a consequence of a less predictable phenology[34,61–65] and more cloud cover (Supplementary Table 7) affecting our ability to map rubber in this region. In addition, we erred on the side of reducing commission errors by applying postclassification masks (as described above). A further explanation is the highly unequal weights of the map classes, with rubber occupying less than 5% of the overall area. Consequently, rubber points mapped as other tree cover led to large area corrections. Finally, the area estimation protocol assumes a completely probabilistic sampling design whereby every point—in accessible and inaccessible locations—had an equal chance to be included. The ground reference data sample design was random but more than 50% of the points were collected in the field (and hence in reasonably accessible areas). This may be a further explanation for the 'over' correction of the rubber class as the correction assumes that every forest point had the same chance to be misclassified as rubber, whether accessible or not. Hence, to err on the side of conservative estimates, we report both area estimates (mapped and sample-based) but concentrate our reports on the smaller one of these figures.

In summary, we developed a new approach, which involves classifying an ESA tree cover baseline map[39] into rubber and other tree cover based on phenology and removing any pixels that are potentially confounded by disturbance using a primary forest mask and a no-disturbance mask, which we generated specifically for this purpose. We also applied a postclassification 5 × 5 pixel majority filter to the resulting map and a minimum patch size threshold of 0.5 ha to reduce pixel-level classification noise and classification artifacts.

### Identifying the deforestation date

We tracked the first historical deforestation date since 1993 for all rubber plantations mapped in 2021. This was done using the LandTrendr spectral-temporal segmentation algorithm[32,75] (a Landsat-based algorithm for the detection of trends in disturbance and recovery). LandTrendr characterizes the history of a Landsat pixel by decomposing the time series into a series of bounded line segments (that is, trends over several years) and identifying the breakpoints between them. These linear segments and breakpoints allow for the detection the greatest pixel-level change (for example, deforestation) and therewith for the identification of the year in which this greatest spectral change occurred (Extended Data Fig. 9).

In this study, we ran LandTrendr GEE API[75] (a JavaScript module developed in Google Earth Engine, https://emapr.github.io/LT-GEE/api.html) using the annual time-series index from USGS Landsat Surface

Reflectance Tier 1 datasets. For hyperparameter settings see Supplementary Table 9. The clouds and cloud shadows were masked using CFMASK[76]. A medoid approach was used to generate the annual composite image. This approach uses the value of a given band that is numerically closest to the median of all the available images for each year. In this study, we used time series of the NBR index (NBR = (NIR − SWIR)/(NIR + SWIR)) from 1993 to 2021 for the temporal segmentation. The deforestation date was identified as the end year of the linear segment with the largest slope (greatest loss). As an extra constraint, we imposed a minimum start NBR value for this linear segment of more than 0.6, thereby reducing the risk of including previously degraded or cleared areas where tree cover was consequently sparser. Any deforestation pixels below this threshold were excluded from our deforestation estimates. We also applied a 3 × 3 pixel majority filter to remove any isolated pixels. To select optimal values for the NBR threshold and the majority filter, we tested combinations of NBR threshold values between 0.51 and 0.61 (in steps of 0.005) with a 3 × 3 and a 5 × 5 pixel majority filter and selected the values that provided maximum overall accuracy. Finally, we excluded pixels with a deforestation date later than 2016 because it takes around 5 years for rubber plantations to be identifiable from the satellite imagery following planting.

As for the rubber map, we evaluated the accuracy of the deforestation date map and calculated estimated area following a standard good practices protocol[33], using all reference sample points (collection described above in the section on 'Mapping the extent of rubber plantations') for which clear deforestation dates could be identified (n = 67). As there were insufficient deforestation reference samples to support a finer temporal classification, we decided to conservatively group the deforestation map into two broad classes: deforestation up to and including 2000 and deforestation between 2001 and 2016. As for rubber, we report all area estimates (mapped and sample-based) to highlight the lowest estimates. Full details of accuracy and area estimates are provided in Supplementary Tables 4–6.

### Deforestation in Key Biodiversity Areas
To explore the potential impacts of rubber and associated deforestation on regional biodiversity we calculated the area of rubber and associated deforestation within KBAs[36]. KBAs are some of the most critical sites for the conservation of species and habitats globally and hence rubber and deforestation in these areas pose a threat to global biodiversity.

### Software
Figure 1 was produced using Colaboratory and Figs. 2 and 3 using Google Sheets.

### Inclusion and ethics
This work is the result of a collaborative partnership between scientists from China and the United Kingdom and includes specialists from inside and outside rubber growing areas. Consideration was given to citation diversity. The study received approval by the Royal Botanic Garden Edinburgh's institutional ethics committee.

### Reporting summary
Further information on research design is available in the Nature Portfolio Reporting Summary linked to this article.

### Data availability
The Earth observation datasets that supported the findings of this study are publicly available (for example, Google Earth Engine data catalogue). The rubber and associated deforestation maps produced here (Fig. 1a,b) are available from https://doi.org/10.5281/zenodo.8425153. They are also available within Google Earth Engine: rubber, https://code.earthengine.google.com/?asset=users/wangyxtina/MapRubberPaper/ rForeRub202122_perc1585DifESAdist5pxPFfinal; associated forest loss, https://code.earthengine.google.com/?asset=users/wangyxtina/MapRubberPaper/rRubber30m202122_deforestationAPI20012016_preNBR600. Source data are provided with this paper.

### Code availability
All code used for this study is available at https://earthengine.googlesource.com/users/wangyxtina/Nature_rubber. Users with a Google Earth Engine account can access the code on https://code.earthengine.google.com/?accept_repo=users/wangyxtina/Nature_rubber.

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

**Acknowledgements** This work was funded by UK Research and Innovation's Global Challenges Research Fund through the Trade, Development and the Environment Hub project (ES/S008160/1) and the Natural Environment Research Council (NE/X016285/1). E.W.-T. was supported by the Natural Environment Research Council NERC-IIASA Collaborative Fellowship (NE/T009306/1) and H.C., Y.S. and J.X. by the Key Research Program of Frontier Sciences, Chinese Academcy of Sciences (QYZDY-SSW-SMC014). The Royal Botanic Garden Edinburgh is supported by the Scottish Government's Rural and Environment Science and Analytical Services Division. We thank C. Ryan, C. Ellis, N. D. Burgess, I. Ahrends, S. Nyquist, O. Cupit, S. Glaser, R. Ebrey and C. Ngo for comments on the manuscript.

**Author contributions** A.A. and Y.W. designed the study. Y.W. devised the computational framework and A.A. and D.Z. provided further support for data analysis. All authors provided feedback and helped shape the analyses. D.Z., H.C., K.H., Y.S., E.W.-T. and J.X. contributed reference ground data. A.A. took the lead in writing the manuscript with input from P.M.H and Y.W. Y.W. took the lead in writing the Methods and producing the figures. All authors reviewed the manuscript.

**Competing interests** The authors declare no competing interests.

**Additional information**
**Correspondence and requests for materials** should be addressed to Yunxia Wang or Antje Ahrends.

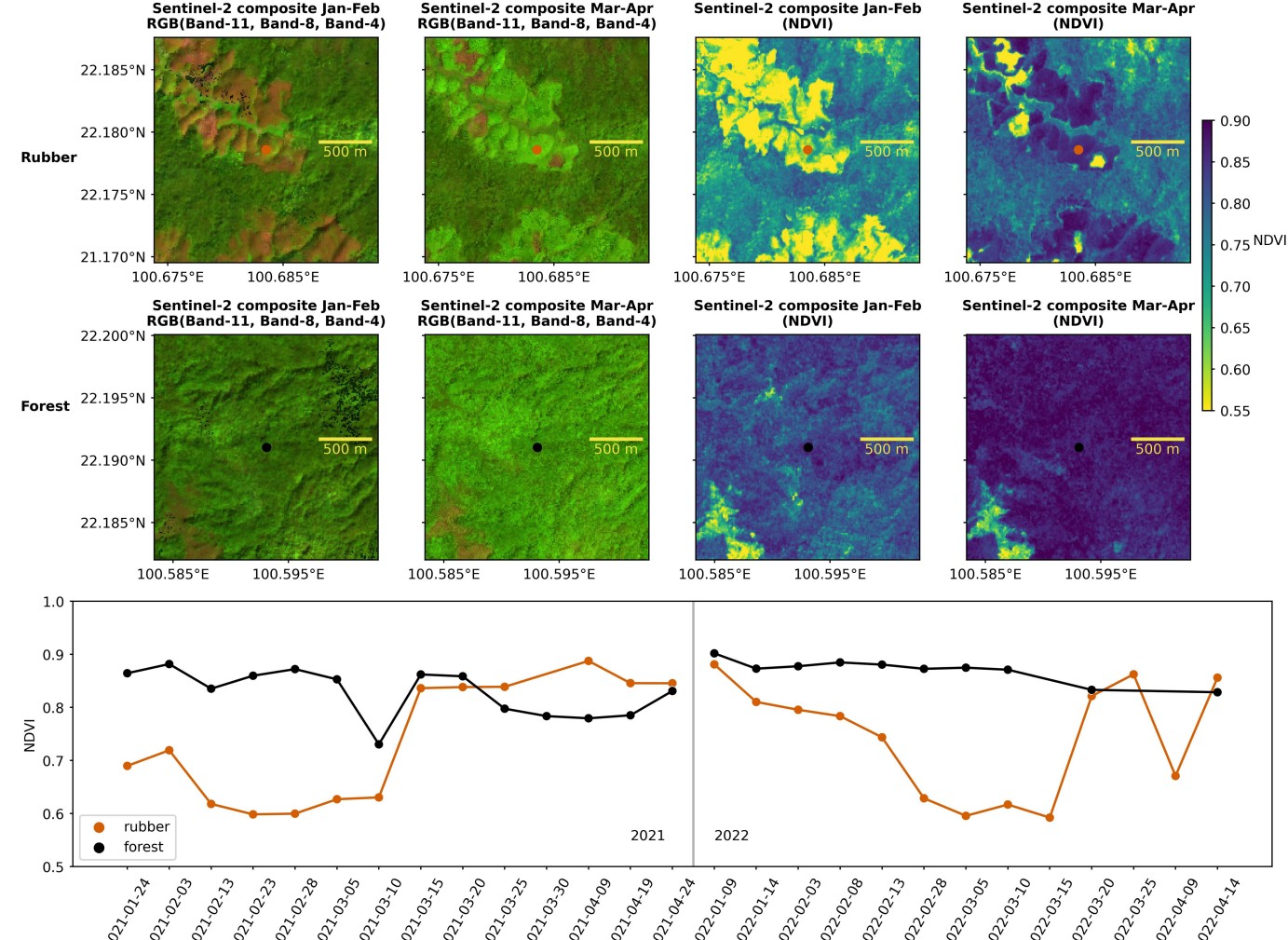

**Extended Data Fig. 1 | Examples of the characteristic spectral signature of rubber and evergreen forests caused by differing phenology in Southeast Asia.** The example pixels for rubber (100.6835 longitude, 22.1786 latitude) and evergreen forest pixels (100.5931 longitude, 22.1910 latitude) shown here are located in Xishuangbanna, China (phenology region A). Rubber has a distinct phenology, shedding leaves in January to February and subsequently refoliating in March and April. Two-year (2021 and 2022) composite image differences between defoliation and refoliation stages were used as inputs for a Random Forest classifier to distinguish rubber and forest. The bottom subplot shows the temporal pattern of the NDVI in January-April 2021 and January-April 2022 (the grey line separates the two years). NDVI: Normalized Difference Vegetation Index $\left(\frac{Band8 - Band4}{Band8 + Band4}\right)$. Images: ESA Sentinel-2. The figure was produced in Colaboratory.

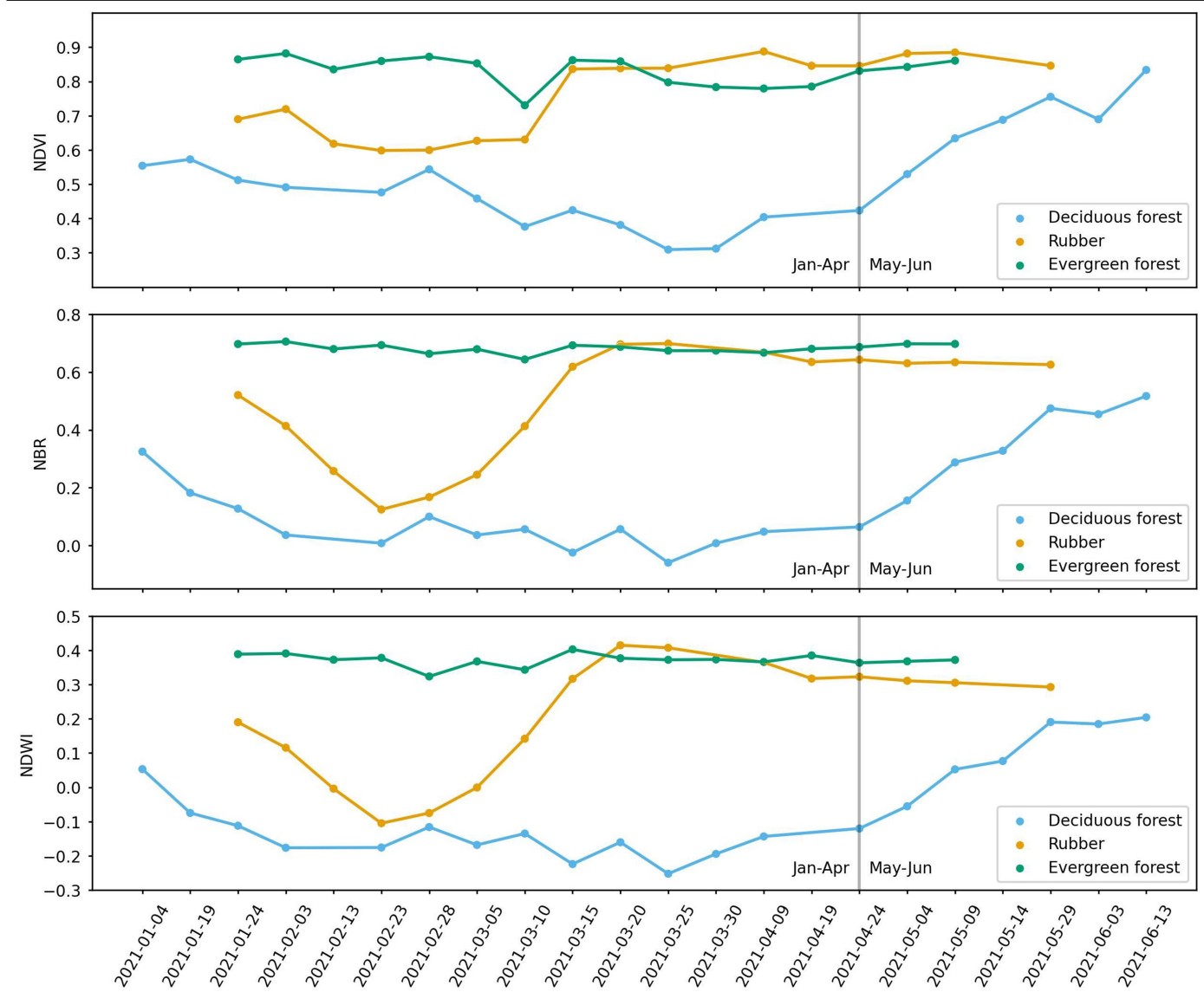

**Extended Data Fig. 2 | Example of differences in Sentinel-2 spectral indices caused by the different phenological responses of rubber, evergreen forest and deciduous forest.** The coordinates for these points are rubber: 100.6835 longitude, 22.1786 latitude; evergreen forest: 100.5931 longitude, 22.1910 latitude; and deciduous forest: 100.7219 longitude, 22.1858 latitude. While the defoliation of deciduous forest lasts until May, rubber defoliation takes place between January and February and the leaves are regained before the onset of the wet season in May. The grey line represents the cut-off date for the composite images used for classifying rubber (when rubber leaves have already flushed but deciduous forest leaves not yet). The figure was produced in Colaboratory. NDVI: Normalized Difference Vegetation Index $\left(\frac{Band8 - Band4}{Band8 + Band4}\right)$. NBR: Normalized Burn Ratio $\left(\frac{Band8 - Band12}{Band8 + Band12}\right)$. NDWI: Normalized Water Index $\left(\frac{Band8 - Band11}{Band8 + Band11}\right)$.

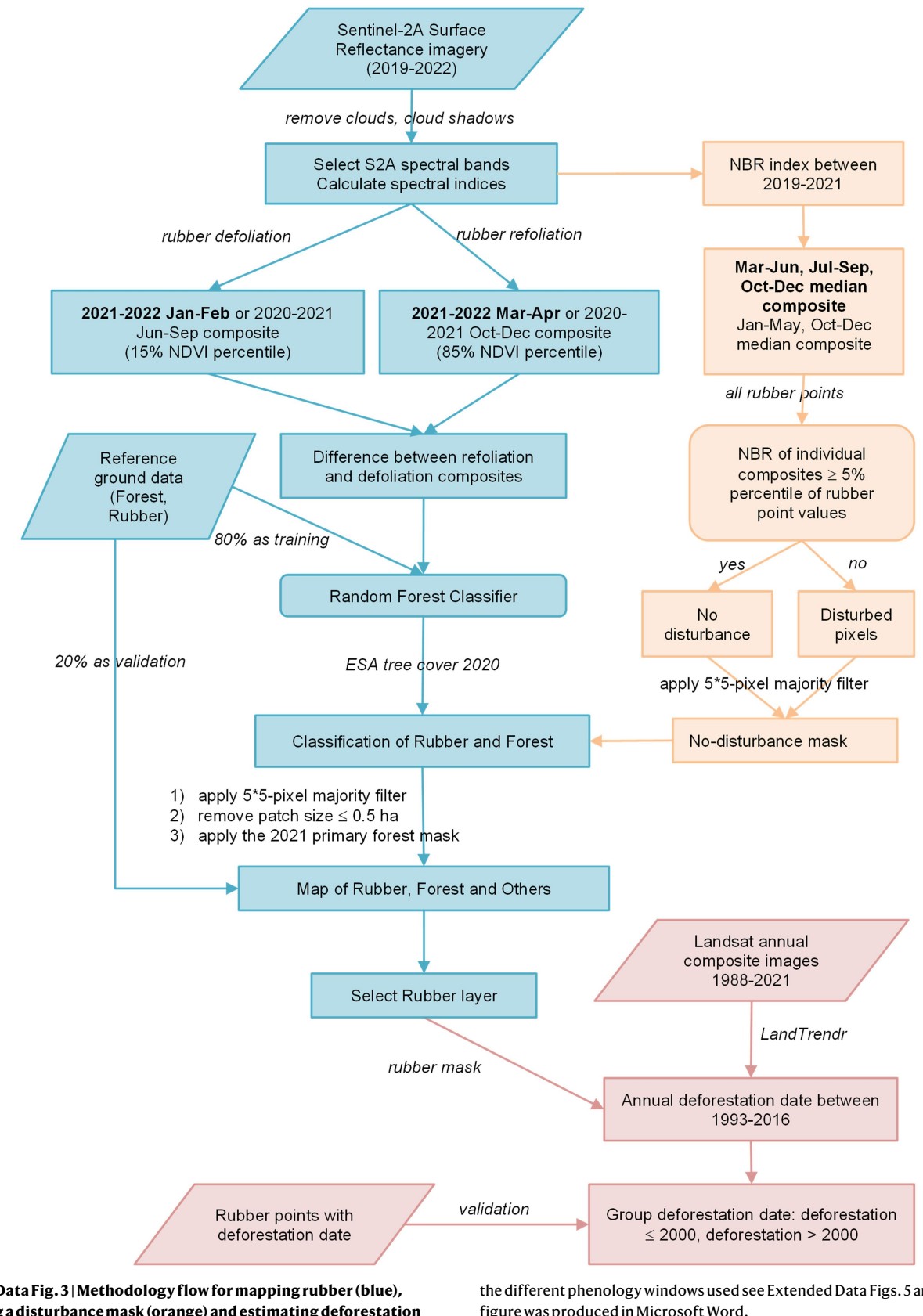

**Extended Data Fig. 3 | Methodology flow for mapping rubber (blue), generating a disturbance mask (orange) and estimating deforestation (red).** All processing was done in Google Earth Engine. For explanations on the different phenology windows used see Extended Data Figs. 5 and 6. The figure was produced in Microsoft Word.

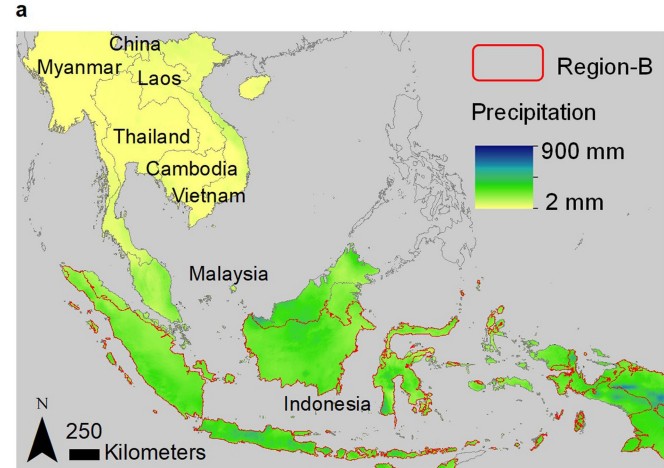

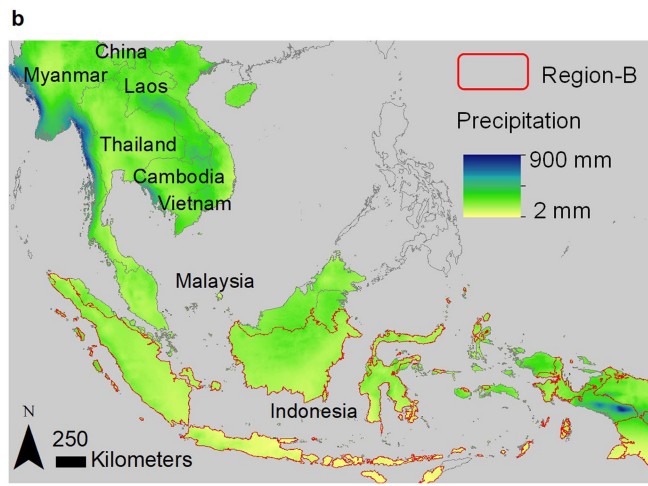

**Extended Data Fig. 4 | Average monthly rainfall during January to February (A) and June to September (B).** Contrary to mainland Southeast Asia, which experiences a distinctive dry season during the northeast monsoon January to February, there is less seasonality in insular Southeast Asia. The areas identified as Region B are generally somewhat drier during June to September when the southwest monsoon brings dry air masses from the Australian continent (*Diercke Weltatlas*. Schulbuchverlage Westermann Schroedel Diesterweg Schoningh Winklers GmbH, 2015). However, the difference is small and, in some areas or years, may never translate into decreased soil moisture (Niu, F., Röll, A., Meijide, A., Hendrayanto & Hölscher, D. Rubber tree transpiration in the lowlands of Sumatra. Ecohydrology 10, doi:10.1002/eco.1882, 2017) and hence not prompt a clear-cut phenological response in rubber. This explains why there are a lot more rubber omission errors in insular Southeast Asia (Supplementary Table 3 and Extended Data Figs. 7 and 8). Rainfall data are from Hengl, T. & Parente, L. (Zenodo: https://doi.org/10.5281/zenodo.6458580, 2022) and administrative boundaries from the Global Administrative Areas database version 1.0. The figure was produced in ESRI ArcMap 10.8.2.

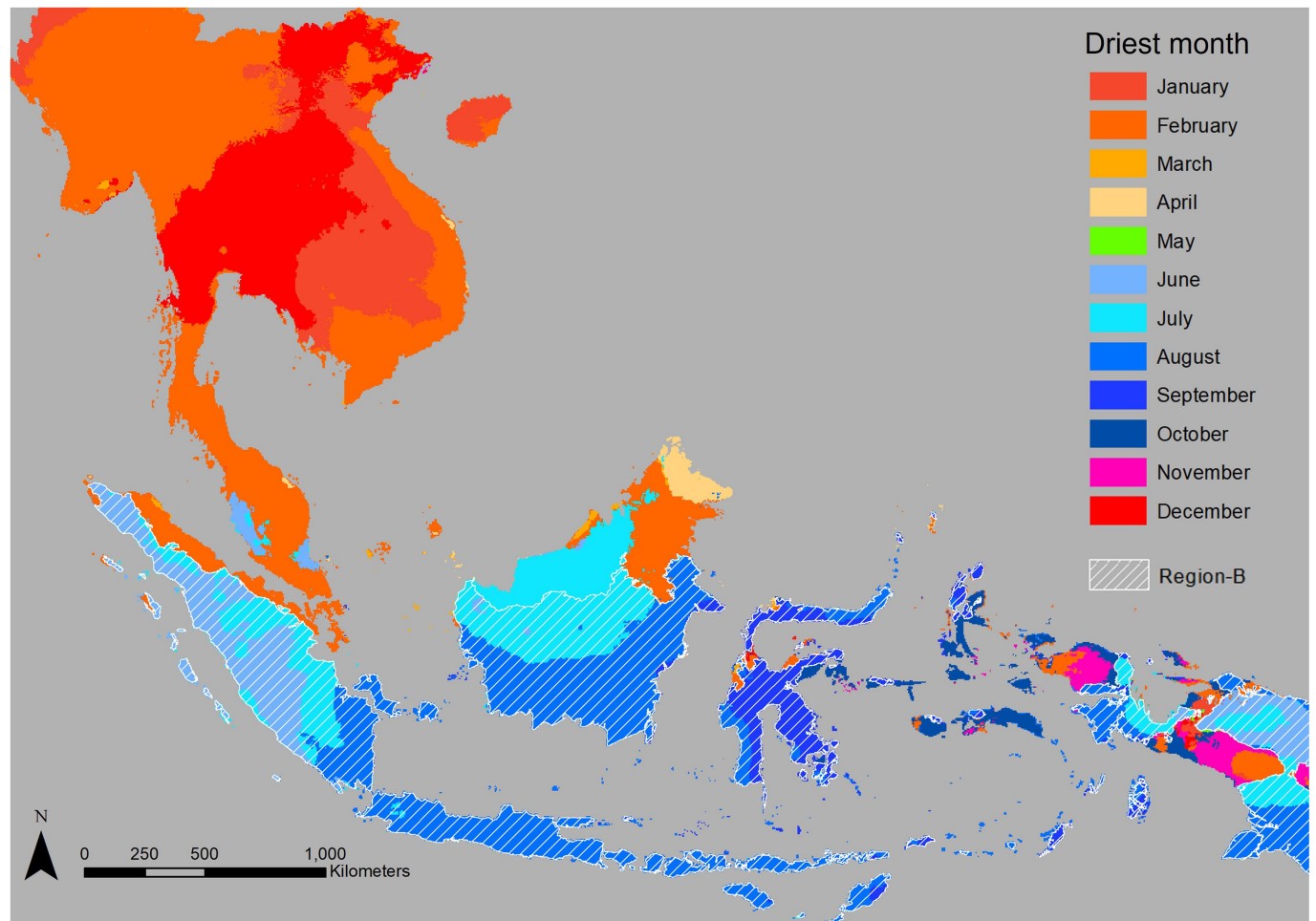

**Extended Data Fig. 5 | Driest month based on 15-year rainfall averages.** To account for the spatial heterogeneity in the onset of the dry wintering season we ran the rubber mapping algorithm separately for two climatic subregions: Region A where rubber defoliation was assumed to occur between January to February and Region B where rubber defoliation was assumed to occur between June to September. Region B was delineated by identifying all pixels ( ~ 1×1 km) in Indonesia where the driest month was either June, July, August or September. All other pixels, including all areas in Malaysia, were assigned to Region A. Owing to heterogenous local topography and wind conditions, rainfall patterns in insular Southeast Asia vary over short distances, in addition to which substantial temporal variation can be present e.g. in the form of the El Niño-Southern Oscillation phenomenon. The division into climatic Regions A and B reflects a trade-off between running the algorithm separately for many small subregions and the need for sufficient ground reference data for robust inferences. In addition, in perhumid areas near the equator (e.g. northern Borneo) this division becomes arbitrary as the lack of seasonality in these areas (Extended Data Fig. 4) precludes a clearly predictable phenological rubber response. Rainfall data are from Hengl, T. & Parente, L. (Zenodo: https://doi. org/10.5281/zenodo.6458580, 2022) and administrative boundaries from the Global Administrative Areas database version 1.0. The figure was produced in ESRI ArcMap 10.8.2.

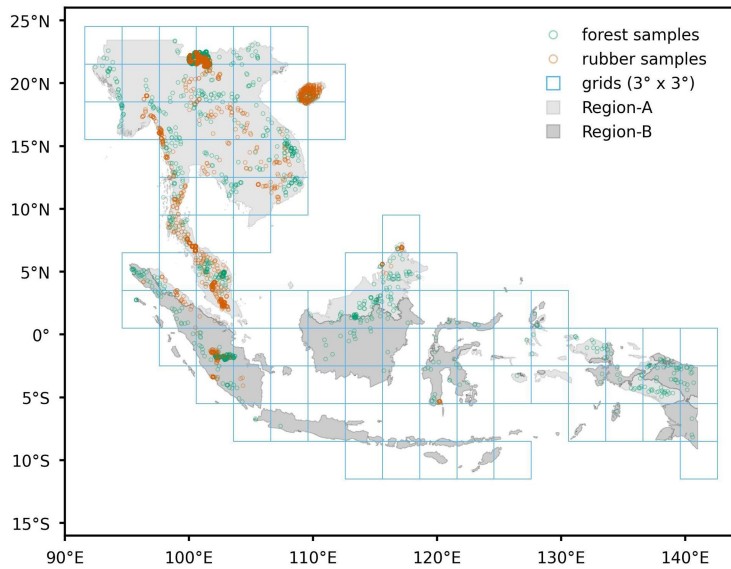

**Extended Data Fig. 6 | Rubber phenology regions, grids and sampling points.** To account for differences in the onset of the dry season we divided the study area into two climatic subregions based on the occurrence of the driest month (Extended Data Fig. 5). Region A: rubber defoliation was assumed to occur between January to February and refoliation between March to April. Region B: rubber defoliation was assumed to occur between June to September and refoliation between October to December. The algorithm was run separately for 3 by 3-degree grid cells (in blue). The forest and rubber reference ground data (open dots; n = 661) were used for training the rubber detection algorithm (80% of the points) and for validating the map (20%). Source of administrative boundaries: The Global Administrative Unit Layers (GAUL) dataset, implemented by FAO within the CountrySTAT and Agricultural Market Information System projects. The figure was produced in Colaboratory.

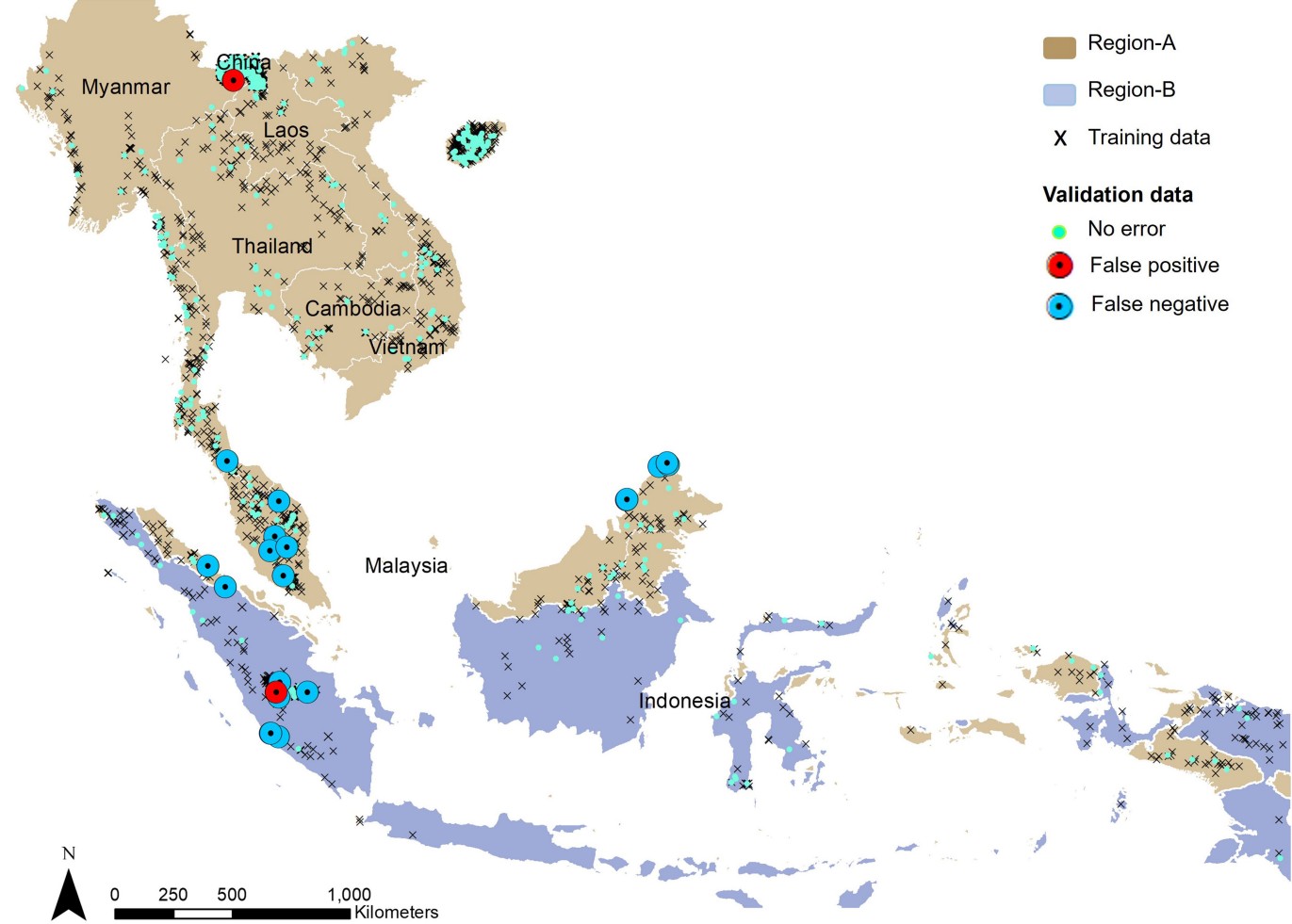

**Extended Data Fig. 7 | Spatial distribution of rubber classification errors.** Of n = 661 validation ground reference points, there were 19 false negatives (of which 18 occurred in Malaysia and Indonesia) and only two false positives (one in Xishuangbanna and one on Sumatra). Source of Administrative boundaries: Global Administrative Areas database version 1.0. The figure was produced in ESRI ArcMap 10.8.2.

**a**

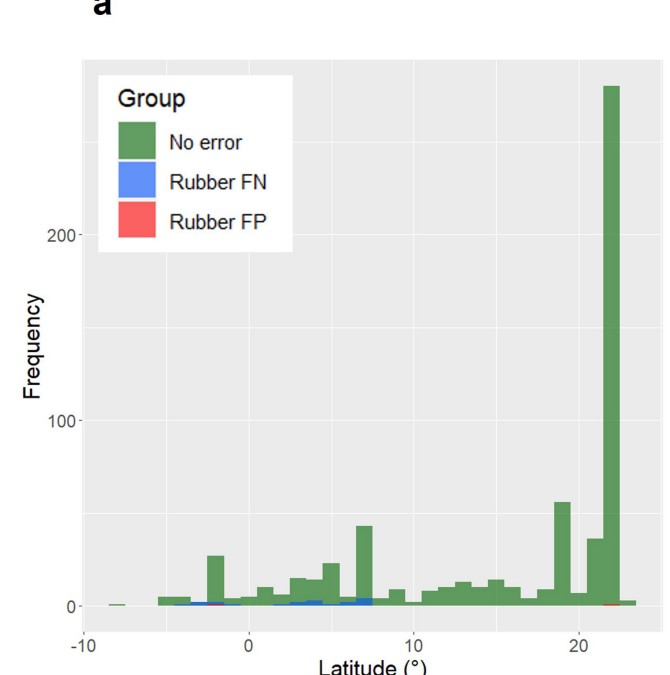
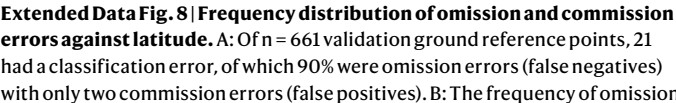

**b**

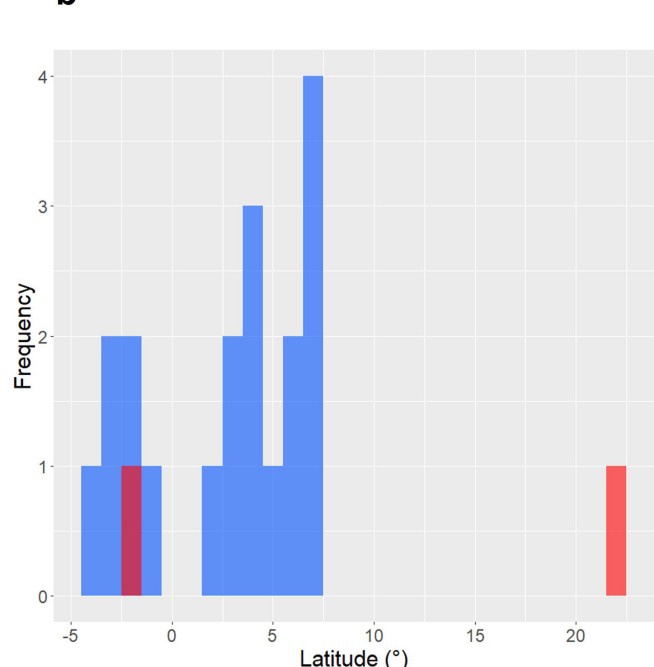

**Extended Data Fig. 8 | Frequency distribution of omission and commission errors against latitude.** A: Of n = 661 validation ground reference points, 21 had a classification error, of which 90% were omission errors (false negatives) with only two commission errors (false positives). B: The frequency of omission errors was highest near the equator. False negatives remained up until c. 7° north. Beyond this point the climate becomes more continental and seasonal (Extended Data Fig. 4) and no more false negatives were found (Extended Data Fig. 7). The figure was produced using R library 'ggplot2'.

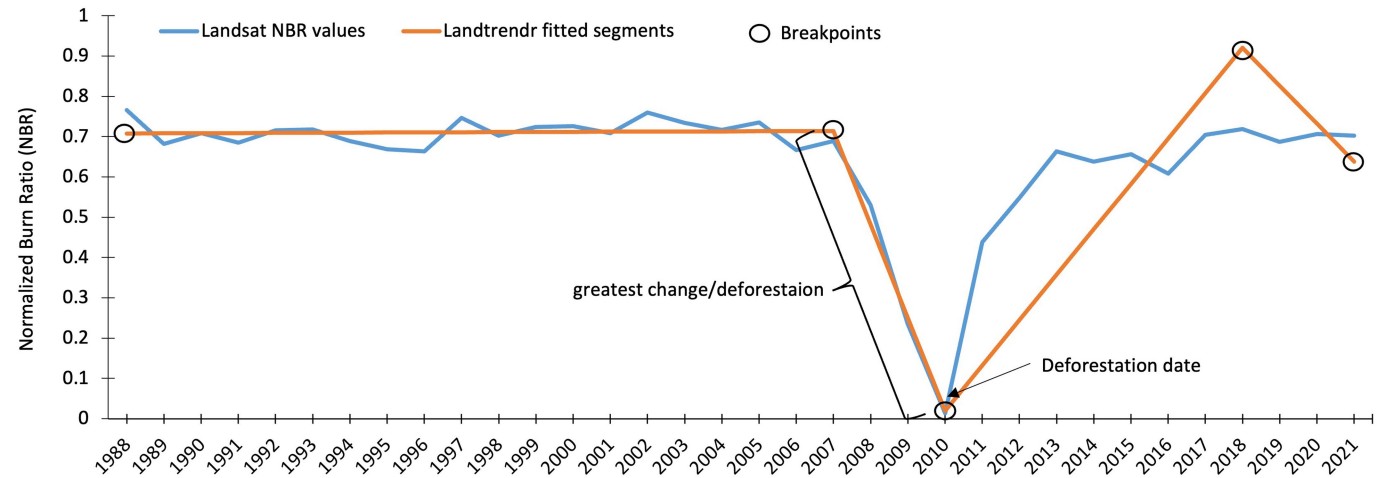

**Extended Data Fig. 9 | Diagram illustrating the LandTrendr segmentation algorithm for detecting historical deforestation using Landsat time series of the Normalized Burn Ratio index.** The example rubber pixel is located in Cambodia (105.4350 longitude, 12.5468 latitude). Further details on the LandTrendr algorithm are available at: https://emapr.github.io/LT-GEE/landtrendr.html. The figure was produced in Microsoft Excel.

# Reporting Summary

## Statistics

For all statistical analyses, confirm that the following items are present in the figure legend, table legend, main text, or Methods section.

| n/a | Confirmed | |
|---|---|---|
| ☐ | ☒ | The exact sample size (*n*) for each experimental group/condition, given as a discrete number and unit of measurement |
| ☐ | ☒ | A statement on whether measurements were taken from distinct samples or whether the same sample was measured repeatedly |
| ☐ | ☒ | The statistical test(s) used AND whether they are one- or two-sided *Only common tests should be described solely by name; describe more complex techniques in the Methods section.* |
| ☐ | ☒ | A description of all covariates tested |
| ☐ | ☒ | A description of any assumptions or corrections, such as tests of normality and adjustment for multiple comparisons |
| ☐ | ☒ | A full description of the statistical parameters including central tendency (e.g. means) or other basic estimates (e.g. regression coefficient) AND variation (e.g. standard deviation) or associated estimates of uncertainty (e.g. confidence intervals) |
| ☒ | ☐ | For null hypothesis testing, the test statistic (e.g. *F*, *t*, *r*) with confidence intervals, effect sizes, degrees of freedom and *P* value noted *Give P values as exact values whenever suitable.* |
| ☒ | ☐ | For Bayesian analysis, information on the choice of priors and Markov chain Monte Carlo settings |
| ☒ | ☐ | For hierarchical and complex designs, identification of the appropriate level for tests and full reporting of outcomes |
| ☒ | ☐ | Estimates of effect sizes (e.g. Cohen's *d*, Pearson's *r*), indicating how they were calculated |

*Our web collection on statistics for biologists contains articles on many of the points above.*

## Software and code

Policy information about availability of computer code

| Data collection | All software used for data collection (Google Earth Engine, Google Earth Pro, and Collect Earth Online) are publicly available. |
|---|---|
| Data analysis | All software used for data analysis (Google Earth Engine, and R 4.2.2) are publicly available and all code is deposited in a public repository: https://earthengine.googlesource.com/users/wangyxtina/Nature_rubber. Users with a Google Earth Engine account can access the code on: https://code.earthengine.google.com/?accept_repo=users/wangyxtina/Nature_rubber |

For manuscripts utilizing custom algorithms or software that are central to the research but not yet described in published literature, software must be made available to editors and reviewers. We strongly encourage code deposition in a community repository (e.g. GitHub). See the Nature Portfolio guidelines for submitting code & software for further information.

## Data

Policy information about availability of data

All manuscripts must include a data availability statement. This statement should provide the following information, where applicable:
- Accession codes, unique identifiers, or web links for publicly available datasets
- A description of any restrictions on data availability
- For clinical datasets or third party data, please ensure that the statement adheres to our policy

The earth observation datasets that supported the findings of this study are publicly available (e.g., Google Earth Engine data catalogue). The rubber and associated deforestation maps produced here (Fig. 1a,b) are available from https://doi.org/10.5281/zenodo.8425153. They are also available from within Google Earth Engine:

Rubber:
https://code.earthengine.google.com/?asset=users/wangyxtina/MapRubberPaper/rForeRub202122_perc1585DifESAdist5pxPFfinal
Associated forest loss:
https://code.earthengine.google.com/?asset=users/wangyxtina/MapRubberPaper/rRubber30m202122_deforestationAPI20012016_preNBR600

# Research involving human participants, their data, or biological material

Policy information about studies with human participants or human data. See also policy information about sex, gender (identity/presentation), and sexual orientation and race, ethnicity and racism.

| | |
|---|---|
| Reporting on sex and gender | n/a |
| Reporting on race, ethnicity, or other socially relevant groupings | n/a |
| Population characteristics | n/a |
| Recruitment | n/a |
| Ethics oversight | n/a |

Note that full information on the approval of the study protocol must also be provided in the manuscript.

# Field-specific reporting

Please select the one below that is the best fit for your research. If you are not sure, read the appropriate sections before making your selection.

☐ Life sciences    ☐ Behavioural & social sciences    ☒ Ecological, evolutionary & environmental sciences

For a reference copy of the document with all sections, see nature.com/documents/nr-reporting-summary-flat.pdf

# Ecological, evolutionary & environmental sciences study design

All studies must disclose on these points even when the disclosure is negative.

| | |
|---|---|
| Study description | Analysis of remotely-sensed data to map rubber in 2021 and to map and quantify associated deforestation |
| Research sample | Reference ground data (in total >3,800 points) based on field observations, augmented with visually interpreted very-high resolution satellite data |
| Sampling strategy | Stratified random. Sample sizes were pre-determined based on a good practices protocol (Olofsson et al. 2014. Good practices for estimating area and assessing accuracy of land change. Remote Sensing of Environment 148, 42-57) |
| Data collection | Of the over 3,800 points, 2,000 were based on randomly sampled reference ground data collected by the World Agroforestry Centre in 2010, covering entire Southeast Asia and consisting of a mix of field data and visually interpreted very-high resolution satellite data. We updated the classification for these points for 2021 following a visual interpretation in Collect Earth Online and Google Earth Pro. The remainder (>1,800 points) were points from randomly sampled field data covering mainland Southeast Asia. |
| Timing and spatial scale | The maps cover 1993 to 2021. The spatial extent encompasses all Southeast Asia at pixel resolutions of 10 and 30 m |
| Data exclusions | No data were excluded |
| Reproducibility | All findings are reproducible and the code to reproduce the findings is available on GitHub. Where parameter choices affected the outcomes, the choices and their effects are clearly outlined in the manuscript. |
| Randomization | Reference ground data were randomly split into training and test data using a random number generator. |
| Blinding | All relevant analyses and data collections were completed and the code was debugged before the results were revealed. |

Did the study involve field work?    ☐ Yes    ☒ No

# Reporting for specific materials, systems and methods

We require information from authors about some types of materials, experimental systems and methods used in many studies. Here, indicate whether each material, system or method listed is relevant to your study. If you are not sure if a list item applies to your research, read the appropriate section before selecting a response.

## Materials & experimental systems

| n/a | Involved in the study |
|-----|----------------------|
| ☒ ☐ | Antibodies |
| ☒ ☐ | Eukaryotic cell lines |
| ☒ ☐ | Palaeontology and archaeology |
| ☒ ☐ | Animals and other organisms |
| ☒ ☐ | Clinical data |
| ☒ ☐ | Dual use research of concern |
| ☒ ☐ | Plants |

## Methods

| n/a | Involved in the study |
|-----|----------------------|
| ☒ ☐ | ChIP-seq |
| ☒ ☐ | Flow cytometry |
| ☒ ☐ | MRI-based neuroimaging |

