## [Peer Review File · Nature]

Manuscript Title: New high-resolution maps show that rubber causes significant deforestation

Reviewer Comments & Author Rebuttals

Reviewer Reports on the Initial Version:

Referees' comments:

Referee #1 (Remarks to the Author):

New high-resolution maps show that rubber causes significant deforestation
Wang et al

This MS presents a detailed remote sensed dataset demonstrating the role of rubber plantation expansion as an important driver of deforestation across SE Asia. Quantifying this impact is important and globally relevant information. Too often the debate around tropical deforestation is focussed on soybeans in the Amazon and palm oil in SE Asia, while ignoring the many other commodities that play a role. The headline figures the authors present demonstrate quite clearly that the importance of crops like rubber is grossly underestimated.

Specific points:

L33-34: This MS doesn't demonstrate any impacts on biodiversity or ecosystem services, and no references are provided. Can the statement be toned down a little? E.g. "...Southeast Asia could be extensive..."

L88: It's worth mentioning specifically whether there are any deciduous tropical forests in the study region that might also have a phenological signal that could be confused with that of rubber.

L97: I couldn't work out how these confidence intervals were generated. The text indicates they are conservative, but the methods don't describe how they were calculated so readers are unable to judge this for themselves. They're also surprisingly wide, given the high level of classification accuracy reported in L101.

Fig. 3: This presents data on the rubber price and introduces some correlations with country-level deforestation rates. The statistics that accompany this are presented only in the legend and there is nothing in the methods related to it, but they should either be removed or improved. It's not appropriate to use Pearson's correlations on time series data and the results of this simplistic analysis are almost certainly misleading. I would encourage the authors to delve more into a formal time series analysis to determine whether there are any significant, time-lagged impacts of rubber price on deforestation. This is important because the authors indicate later on in the MS that future price increases could drive additional deforestation (L272-274; see additional comment below), but that assumption has not been tested.

L247: Are the land balance models a zero-sum model, meaning that the increased importance of rubber demonstrated in these new data must mean a reduction in the assumed importance of other crop types? If so, what are those other crops that might now be de-emphasised in discussions of tropical deforestation?

L255: There's an analogous process in the Amazon where high value soya plantations can drive low value cattle farming into new locations. The point being made here would be stronger if it's

cast as a generalised phenomenon, with rubber being a specific example.

L272-274: What drove the peak in rubber price in 2011, and is it reasonable to expect such a peak to occur again? Why do you assume the price will go up again?

L498: Why resample down to 10 m resolution when the native resolution of most of the input data are 20 m?

Referee #2 (Remarks to the Author):

This is a very timely paper with important findings. It uses advances in remote sensing to prepare a much updated estimate of rubber cultivation area and impact on key biodiversity areas.

The paper is well written, uses a solid methodology (with its inherent limitations well discussed) and the presentation of the results is appropriate. It has important policy implications and there is a need to grow consumer awareness on the relatively large impacts of rubber.

As I strongly believe that a reviewer should not ask for many comments and modifications if a paper is strong, I have refrained from making many comments. Just a few issues would need to be resolved:

- update the information on the European deforestation-free commodities: I believe the current version does include rubber (fortunately)

- there is a bit too much repetition between intro and discussion of some (important items) like the smallholder benefits. Carefully reading those sections again will help to avoid too much repetition of the same points and may even condense the paper a bit more

- rubber areas in Africa are fastly growing, and the EU is a prime importer from these. It seems that some of your numbers are a bit outdated on that context. If you can find more recent numbers it would be great (otherwise just tell us that this is indeed the most recent).

- the maps: would be nice to make these a bit more visually attractive in terms of layout and display, maybe with a zoom in to a key production area to better understand the variation in spatial variability in the landscape

Congratulations with a very nice paper

Peter Verburg

Referee #3 (Remarks to the Author):

The study provides spatially explicit data of the rubber tree cultivation area in Southeast Asia. It is based on a remote sensing approach and encompasses a region that likely represents 90% of the global rubber production volume. Furthermore, the rubber-related deforestation history was analysed for different periods of time.

The rubber area was estimated at 14.5 million hectares with a considerable uncertainty (5.6 - 23.4 95% CI). This estimate is not very different derived from FAO statistics. The great advantage of the present study is, that it is spatially explicit, comes at a high resolution and provides confidence intervals. This is highly appreciated.

The associated area of deforestation between 1993 and 2016 was estimated at 3.76 million ha (almost 75% of this after 2001). The rubber deforestation rates used by policy makers thus far are significantly lower.

I find this a very valuable study.

I also have some remarks

1) The approach is based on rubber leaf phenology. Unfortunately, very little is known or understood about rubber tree's leaf phenology. The study uses a classification into regions, where rubber defoliation occurs during January-February (region-A), and where rubber defoliation occurs in June-September (region-B). The Extended Data Fig.2 shows these rubber phenology regions.

The A and B, black or white ... approach is of course very simple, not to say simplistic. I miss a discussion and further exploration of this topic.

Even on land, there is often a suggested abrupt change from presumed leaf shedding in June-September and January-February; e.g. on the islands Sumatra, Borneo and Papua-New Guinea. Along this very long dividing line and in an unknown extent of its corridor, there could be a huge uncertainty in classification.

2) The rubber signature, based on leaf shedding, was compared with evergreen rainforest. 1,816 evergreen forest points from ground-truthed points were analysed.

I wonder: There are large areas with natural deciduous or semi-deciduous natural forests in the study region, in particular in Mainland Southeast Asia. How were points analysed that fell into such regions?

3) A Random Forest machine learning classification was applied 80% sample points for training and 20% for validation of the final rubber map. The performance statistics should be communicated more comprehensively.

4) 'Fig.3. Rates of rubber deforestation in some countries were strongly correlated with the global rubber price'. This is an interesting observation. Certainly, also other variables will correlate with deforestation. Without further exploration, the communicated finding might be misleading.

5) L 231 and following. An additional uncertainty arises from the formerly in Indonesia widespread mixed species cultivation system with rubber, the so-called jungle rubber. I would guess, this type of rubber cultivation is not detectable by this remote sensing approach. I mean today, its role might be neglectable but in the past?

Van Noordwijk, M., Tata, H. L., Xu, J., Dewi, S., & Minang, P. A. (2012). Segregate or integrate for multifunctionality and sustained change through rubber-based agroforestry in Indonesia and China. In P. K. R. Nair, & D. Garrity (Eds.), *Agroforestry—The future of global land use* (pp. 69–104). Netherlands Pp: Springer. ISBN: <https://doi.org/info:x-wiley/isbn/9789400746756>.

6) Key biodiversity areas. There is not a word on how rubber cultivation might affect biodiversity. The study by Grass et al. may be helpful in this context.

Grass, I., C. Kubitz, V.V. Krishna, M.D. Corre, O. Mußhoff, J. Drescher, K. Rembold, P. Pütz, E. Sulpin Ariyanti, A. Barnes, U. Brose, B. Brümmer, D. Buchori, R. Daniel, K. Darras, H. Faust, J. Hein, N. Hennings, P. Hidayat, D. Hölscher, M. Jochum, A. Knohl, M. Kotowska, V. Krashevskaya, H. Kreft, C. Leuschner, N. Jun Lobite, R. Panjaitan, A. Polle, A. Potapov, M. Qaim, A. Röhl, S. Scheu, D. Schneider, A. Tjoa, T. Tschardt & E. Veldkamp, 2020. Trade-offs between multifunctionality and profit in tropical smallholder landscapes. *Nature Communications* 11, 1186.

L73 '... with palm oil and soy accounting for seven to eight times more deforestation than rubber' To me, the wording appears not entirely clear. Does that mean palm oil and soy combined? Or each of them palm oil and soy?

L74 I personally don't find the nutmeg comparison very convincing

L75 I feel: one 'the' too much

L621 available comes twice

L 641 Not correctly cited. Hendaryanto is missing

Fig 2. I don't find the line connecting different countries appropriate. This would suggest, that there might be values somewhere in-between the countries.

Referee #4 (Remarks to the Author):

Review: New high-resolution maps show that rubber causes significant deforestation

This study proposes a new method to detect and map deforestation associated with rubber tree plantations across Southeast Asia. The authors point out that rubber crop deforestation is estimated through modeling and extrapolation, leading to high uncertainty. To overcome it, satellite observation data was processed in the cloud computing Google Earth Engine platform. Their results indicate that "deforestation for rubber is two to threefold higher than suggested by figures currently." 3.7 million hectares of forest are suggested to be cleared for rubber plantation in Southeast Asia, according to the study's results, 27% (i.e., 1 million hectares) in key biodiversity areas. This study deserves publication, but major issues must be improved before acceptance. First, several unclear steps in the remote sensing approach must be clarified. I provide a detailed list below, but the most critical is the lack of area adjustment using the 'good practices' for area estimation proposed by Olofsson et al. (2014). The whole point of this study is to reduce the uncertainty of the model-extrapolation approach, but it seems no area uncertainty from the remote sensing estimates is presented (properly). Secondly, the sampling design for assessing the rubber deforestation map is also unclear. Third, it seems that uncertainty varied across the study area (mainly due to the availability of satellite observation). This must be quantified and presented because there are policy implications in areas where rubber deforestation cannot be estimated accurately. Finally, there are missing details of the methodology used to detect and map rubber deforestation (see more information below). I could not assess the map results because the scripts and maps were not shared - I suggest the authors include them in the new version of the manuscript. See below a list of other major and minor issues that must be addressed.

Major and Minor Issues:

Line 26: Please, define high-resolution since there is no standard for it. Landsat is called 'medium spatial resolution' (see: <https://www.frontiersin.org/articles/10.3389/frsen.2022.894571/full>), and Sentinel-2 fits this class of Earth Observation data.

Line 31: Please, put the rubber deforestation estimates in a regional and global context. What percent does it represent relative to the total deforestation in the study area?

Lines 34-36: I suggest the authors be more propositive and point to domestic and global policies. The conclusion statement in the Abstract is quite general. Which ongoing supply chain deforestation-free forum could it fit? What is the implication of rubber deforestation on carbon emission estimates? Are countries neglecting it in their CO2 emission reports?

Line 87-89: It has been said previously that rubber plantation is spectrally indistinguishable from tropical forests. What makes it possible now? Any specific Sentinel-2 spectral band can do the job?

Line 91-93: The statement "We track the deforestation..." shows up abruptly. How does it link with the Sentinel-2 analysis?

Line 93-96: Please, define 'deforestation' in the context of this study. It is unclear. This whole sentence of this last paragraph is confusing.

Line 101: Please, include the user's accuracy. Explain briefly which statistical protocol to assess the accuracy was used. Also, explain how the rubber deforestation area was estimated. Simply counting pixels lead to a biased estimate which can be adjusted using the error matrix (see: Olofsson et al., 2004).

Figure 1. Please replace '500 m resolution' with '500 m pixel size' for clarity and in '500 m'. The maps are blurred, lack a scale north arrow, and are too small. Include important geographic location and ocean name. The cartographic design is poor. I suggest expanding the map to fit one page.

Line 121: Please, clarify how the NBR index was used in this part of the manuscript.

Figure 3. Why did you use curved lines in this graph? Did you fit a model to estimate the area? Do not smooth the statistics for aesthetic purposes. What is the uncertainty of the estimate per country? I suggest to breakdown the accuracy assessment per country since the results vary across the countries due to satellite data availability, especially in the study area.

Line 216: What is the uncertainty of the ESA tree cover map? This might affect the overall result.

Line 218: Cloud cover is the most deleterious problem for assessing land cover change in Southeast Asia with optical remote sensing, including Landsat and Sentinel-2. This needs to be treated adequately in this study.

Line 220: What is the percent of the ground observed area?

Line 221: What do you mean by 'potentially' more accurate? The map is or isn't more accurate. Disaggregate your reference data used to assess the map accuracy and quantify how the map accuracy varies across the countries).

Line 493: Please, justify the seven vegetation indices used. NDWI, NBR, and MNBR are not vegetation indices. So, why were they used? Isn't their redundancy amongst the vegetation indices used? I believe so. What is the rationale for selecting them?

Method: Please, include a detailed workflow of the processes and details about the data sets used and the parameters of the main algorithms.

Line 509: Examples of phenology spectral curves of the rubber and forest trees must be provided for representative sites.

Line 525-527: Apparently, only NDV was used to classify rubber and non-rubber trees. Please, clarify it and indicate how the other spectral indices were applied.

Line 535: Please, replace 'ground-true points' with 'reference ground data.'

Line 537-539: Please, provide more details about the sampling design. How did you get the 1,000 points for each class?

Line 540: Please, provide examples of images used in the visual interpretation of rubber and forest and more detail about the visual interpretation protocol. How many image interpreters? How was

the reference data assessed? What are the labeling criteria?

Line 547: The high-resolution imagery from the Google Earth base map does not have the image acquisition date. How did it affect the interpretation? Justify the reference data splitting threshold.

Line 571: The 5x5-pixel majority filter changes the border, which affects the area estimate. Please, assess the effect of this filter on the area estimate by comparing the area with and without the filtering process.

Line 592: Please, add a figure showing examples of deforestation breakpoints, including the minimum threshold imposed. By the way, the rationale of this threshold needs to be clarified, and the figure can help explain it.

Line 597: What is the implication of excluding deforestation later than 2016? Does it imply that the algorithm produces maps of rubber deforestation five years out of date?

Data and code availability: Provide the links to access the dataset and map results and details of Google Earth Engine scripts to access the satellite and geospatial data sets used in this research.

It was impossible to assess these data sets since the links were not shared.

Author Rebuttals to Initial Comments:

Referees' comments:

Referee #1 (Remarks to the Author):

New high-resolution maps show that rubber causes significant deforestation
Wang et al

This MS presents a detailed remote sensed dataset demonstrating the role of rubber plantation expansion as an important driver of deforestation across SE Asia. Quantifying this impact is important and globally relevant information. Too often the debate around tropical deforestation is focussed on soybeans in the Amazon and palm oil in SE Asia, while ignoring the many other commodities that play a role. The headline figures the authors present demonstrate quite clearly that the importance of crops like rubber is grossly underestimated.

Specific points:

- (1) L33-34: This MS doesn't demonstrate any impacts on biodiversity or ecosystem services, and no references are provided. Can the statement be toned down a little? E.g. "...Southeast Asia could be extensive..."

Statement amended as advised.

- (2) L88: It's worth mentioning specifically whether there are any deciduous tropical forests in the study region that might also have a phenological signal that could be confused with that of rubber.

Thank you, that is worth mentioning. We have now made clear that our methods allow distinguishing rubber from both evergreen and deciduous forest (lines 97-101). We have also added more details (Methods lines 591-599, Extended Data Figs. 1 and 2) to explain how the distinction is achieved: In a nutshell, the spectral signature between these two systems differs with respect to the timing of leaf regrowth. Rubber refoiliates just before the onset of the wet season (March-April) while for deciduous forest and other deciduous tree crops in mainland Southeast Asia leaf regrowth coincides with the onset of the wet season in May. Hence, we minimise the potential inclusion of deciduous forest and other tree crops by focussing the classification on spectral differences between January-February and March-April.

(See also point 2 in response to reviewer 3.)

- (3) L97: I couldn't work out how these confidence intervals were generated. The text indicates they are conservative, but the methods don't describe how they were calculated so readers are unable to judge this for themselves. They're also surprisingly wide, given the high level of classification accuracy reported in L101.

Thank you, this is a good point. We have now made much clearer how the confidence intervals were generated (Caption Table 1, Results lines 118, Methods lines 665-666, 717-718). We have also

included all calculations for estimating accuracy and confidence intervals into the Supplementary Material (Extended Data Tables 1-6). The originally wide confidence intervals were in fact overly conservative. This was due to the inclusion of a third but redundant map class ('other') for which we had very few reference sample points). Removing that class has reduced the size of the confidence intervals and is also more consistent with our approach (distinguishing rubber from any other types of tree cover).

The confidence intervals and all other accuracy estimates were based on the good practices approach described by Olofsson et al. 2014. *Remote Sensing of Environment* 148, 42-57.

(See also points 1 and 11 in response to reviewer 4.)

(4) Fig. 3: This presents data on the rubber price and introduces some correlations with country-level deforestation rates. The statistics that accompany this are presented only in the legend and there is nothing in the methods related to it, but they should either be removed or improved. It's not appropriate to use Pearson's correlations on time series data and the results of this simplistic analysis are almost certainly misleading. I would encourage the authors to delve more into a formal time series analysis to determine whether there are any significant, time-lagged impacts of rubber price on deforestation. This is important because the authors indicate later on in the MS that future price increases could drive additional deforestation (L272-274; see additional comment below), but that assumption has not been tested.

We agree and we have now removed the simplistic correlations (see also point 4 in response to reviewer 3.) Instead, we point out that rates of rubber expansion and associated deforestation are driven by the decisions of millions of actors, influenced by complex and interlinked drivers such as national policies and subsidies, prices for other crops, and the availability of extension services and infrastructure (caption Fig. 3). We still point out that rates of deforestation in Cambodia and Vietnam have increased in line with the recent surge of the global rubber price, but we are no longer drawing a simplistic correlation and we have also toned down the associated statement in the discussion (see response to comment 7 below).

(5) L247: Are the land balance models a zero-sum model, meaning that the increased importance of rubber demonstrated in these new data must mean a reduction in the assumed importance of other crop types? If so, what are those other crops that might now be de-emphasised in discussions of tropical deforestation?

This is a good point and we have now added a sentence on this to the manuscript on lines 277-280. The land balance model has a lot of excess deforestation that cannot be explained by crop expansion (over 50 million ha in the first version of this dataset covering 2005-2017). Thus, rubber will simply account for some of the unattributed deforestation rather than de-emphasising other crops. We have contacted the authors of the land balance model who confirmed that this is the case.

(6) L255: There's an analogous process in the Amazon where high value soya plantations can drive low value cattle farming into new locations. The point being made here would be stronger if it's cast as a generalised phenomenon, with rubber being a specific example.

Thank you. We have considered this but decided to err on the side of caution with that comparison as although the pattern/outcome is similar, the economic forces driving these processes are subtly

different – in that putting cattle onto land can be strategic and speculative and involves clearing the land and establishing a claim ahead of putting the more profitable soy onto it. In the case of smallholder driven rubber the shift is more due to a comparative change in fortunes of oil palm versus rubber and the price volatility. Thus, the transition to rubber does not involve the speculation for eventual oil palm production.

Grass, I. *et al.* Trade-offs between multifunctionality and profit in tropical smallholder landscapes. *Nat Commun* **11**, 1186, doi:10.1038/s41467-020-15013-5 (2020).

(7) L272-274: What drove the peak in rubber price in 2011, and is it reasonable to expect such a peak to occur again? Why do you assume the price will go up again?

Thank you. We have worded this now more carefully (“...a problem that could increase if rubber prices rise again” instead of “...a problem that *is likely to* increase *when* rubber prices rise again”. (Also see response to comment 4 above).

Indeed, although price bubbles are characteristic for natural rubber, and although industry analysts generally expect the price to rise again, the timing and nature of future price increases is of course unpredictable. According to industry experts, the price peak in 2011 was mainly driven by the expanding automotive industry in Southeast Asia; in particular in China – the globally largest consumer of natural rubber. Since the crash the price has been remained low, fluctuating around supply demand elasticity: for example, following the devastating 2017 floods in southern Thailand and the destruction of many plantations, the price temporarily doubled, but during the COVID-19 pandemic and a global semiconductor shortage the price fell due to temporarily reduced vehicle sales.

According to the 2022 Industry Outlooks published by the International Rubber Study Group the demand for natural rubber is likely to pick up again between now and 2030, but with many rubber farmers currently waiting for price increases before tapping again and many unsold stocks it could indeed take several more years for increased demand to manifest in increased prices again.

(8) L498: Why resample down to 10 m resolution when the native resolution of most of the input data are 20 m?

Thank you, this is a good point that deserves justification. We have now added a sentence to the Methods to explain this choice (lines 576-579). Working with a 10 m instead of a 20 m resolution allowed us to take advantage of the high resolution of key bands (e.g., the NDVI component bands B4 and B8). With smallholder plantations often being below 1 ha in size they may otherwise be missed (as we also apply a 5 by 5-pixel majority filter).

Referee #2 (Remarks to the Author):

This is a very timely paper with important findings. It uses advances in remote sensing to prepare a much updated estimate of rubber cultivation area and impact on key biodiversity areas.

The paper is well written, uses a solid methodology (with its inherent limitations well discussed) and the presentation of the results is appropriate. It has important policy implications and there is a need to grow consumer awareness on the relatively large impacts of rubber.

As I strongly believe that a reviewer should not ask for many comments and modifications if a paper is strong, I have refrained from making many comments. Just a few issues would need to be resolved:

(1) -update the information on the European deforestation-free commodities: I believe the current version does include rubber (fortunately)

Thank you very much for these helpful comments. We have now updated the information on the EUDR and added a note that a pre-print version of our manuscript was part of the evidence contributing to the trialogue in December 2022, which adopted a last-minute decision to include rubber.

(2) -there is a bit too much repetition between intro and discussion of some (important items) like the smallholder benefits. Carefully reading those sections again will help to avoid too much repetition of the same points and may even condense the paper a bit more

Thank you, we have now amended parts of the Introduction and Discussion. Some repetition on important items such as smallholders inevitably remains as we would like to highlight the critical importance of incoming legislation not marginalising smallholders. In recent discussions with industry stakeholders over how our maps may be able to guide compliance monitoring this point has repeatedly emerged, with actors concerned that a response could be for companies affected by the EUDR to exclusively source from large industrial plantations thereby minimising problems associated with traceability.

(3) -rubber areas in Africa are fastly growing, and the EU is a prime importer from these. It seems that some of your numbers are a bit outdated on that context. If you can find more recent numbers it would be great (otherwise just tell us that this is indeed the most recent).

We have now taken out the numbers related to EU imports as indeed they are based on data sources that may now be outdated given the rapid rise of Africa as a production region. We are in the process of mapping rubber in Africa but due to the relative data scarcity this work is still at very early stages.

(4) -the maps: would be nice to make these a bit more visually attractive in terms of layout and display, maybe with a zoom in to a key production area to better understand the variation in spatial variability in the landscape

Thank you, we have improved the figure in line with these suggestions. (See also point 12 in response to reviewer 4).

Congratulations with a very nice paper
Peter Verburg

Referee #3 (Remarks to the Author):

The study provides spatially explicit data of the rubber tree cultivation area in Southeast Asia. It is based on a remote sensing approach and encompasses a region that likely represents 90% of the

global rubber production volume. Furthermore, the rubber-related deforestation history was analysed for different periods of time.

The rubber area was estimated at 14.5 million hectares with a considerable uncertainty (5.6 - 23.4 95% CI). This estimate is not very different derived from FAO statistics. The great advantage of the present study is, that it is spatially explicit, comes at a high resolution and provides confidence intervals. This is highly appreciated.

The associated area of deforestation between 1993 and 2016 was estimated at 3.76 million ha (almost 75% of this after 2001). The rubber deforestation rates used by policy makers thus far are significantly lower.

I find this a very valuable study.

I also have some remarks

(1) The approach is based on rubber leaf phenology. Unfortunately, very little is known or understood about rubber tree's leaf phenology. The study uses a classification into regions, where rubber defoliation occurs during January-February (region-A), and where rubber defoliation occurs in June-September (region-B). The Extended Data Fig.2 shows these rubber phenology regions. The A and B, black or white ... approach is of course very simple, not to say simplistic. I miss a discussion and further exploration of this topic. Even on land, there is often a suggested abrupt change from presumed leaf shedding in June-September and January-February; e.g. on the islands Sumatra, Borneo and Papua-New Guinea. Along this very long dividing line and in an unknown extent of its corridor, there could be a huge uncertainty in classification.

Thank you, this is an important point. We have now added a discussion on this (Discussion lines 226-232, Methods lines 599-604). In addition, we are now explicitly quantifying accuracy by region (mainland versus insular Southeast Asia). Our estimates for mainland Southeast Asia are considerably more precise and accurate than for insular Southeast Asia where sample data suggest potential large omissions. This will indeed be due to the unpredictable phenology of rubber in these areas. We are now making that clear and we also state that our maps are likely to be highly conservative with sample data suggesting larger omissions in insular Southeast Asia (see Results lines 121-129, Methods lines 666-670, caption Fig. 1, caption Table 1, Extended Data Tables 2 and 3).

(See also point 3 in response to reviewer 4).

(2) The rubber signature, based on leaf shedding, was compared with evergreen rainforest. 1,816 evergreen forest points from ground-truthed points were analysed. I wonder: There are large areas with natural deciduous or semi-deciduous natural forests in the study region, in particular in Mainland Southeast Asia. How were points analysed that fell into such regions?

Thank you, this requires clarification in the manuscript. There are indeed large areas with natural deciduous and semi-deciduous forest in the study region. We have now made clearer in the manuscript that our methods do allow us to distinguish rubber from deciduous forest (Introduction lines 97-101 and Methods lines 591-599). We have also added two figures to the Supplementary

Material (Extended Data Figs. 1 and 2) to show how this is achieved: The spectral signature between these two systems differs with respect to the timing of leaf regrowth. Rubber refoiliates just before the onset of the wet season (March-April) while in deciduous forest and other deciduous tree crops in mainland Southeast Asia refoiliation mainly occurs with the onset of the wet season in May. Hence, we minimise the potential inclusion of deciduous forest and other tree crops by focussing the classification on spectral differences between January-February and March-April.

(See also point 2 in response to reviewer 1.)

(3) A Random Forest machine learning classification was applied 80% sample points for training and 20% for validation of the final rubber map. The performance statistics should be communicated more comprehensively.

Yes. We have added a lot more detail on the validation of our rubber maps (Extended Data Tables 1-3). In addition, we have added two tables to the Supplementary Material (Extended Data Tables 8 and 9) which provide more detail on the Random Forest classifier (hyperparameter settings and relative variable importance in deriving the classification).

(See also points 1, 11 and 19 in response to reviewer 4.)

(4) ‘Fig.3. Rates of rubber deforestation in some countries were strongly correlated with the global rubber price’. This is an interesting observation. Certainly, also other variables will correlate with deforestation. Without further exploration, the communicated finding might be misleading.

We agree and we have now removed the simplistic correlations. Instead, we point out that rates of rubber expansion and associated deforestation are driven by the decisions of millions of actors, influenced by complex and interlinked drivers such as national policies and subsidies, prices for other crops, and the availability of extension services and infrastructure (caption Fig. 3). We still point out that rates of deforestation in Cambodia and Vietnam have increased in line with the recent surge of the global rubber price, but we are no longer drawing a simplistic correlation.

(See also point 4 in response to reviewer 1).

(5) L 231 and following. An additional uncertainty arises from the formerly in Indonesia widespread mixed species cultivation system with rubber, the so-called jungle rubber. I would guess, this type of rubber cultivation is not detectable by this remote sensing approach. I mean today, its role might be neglectable but in the past?

Van Noordwijk, M., Tata, H. L., Xu, J., Dewi, S., & Minang, P. A. (2012). Segregate or integrate for multifunctionality and sustained change through rubber-based agroforestry in Indonesia and China. In P. K. R. Nair, & D. Garrity (Eds.), *Agroforestry—The future of global land use* (pp. 69–104). Netherlands Pp: Springer. ISBN: <https://doi.org/info:x-wiley/isbn/9789400746756>.

Thank you, that is an important point, which we have now added to the Discussion (lines 236-237), along with the reference.

(6) Key biodiversity areas. There is not a word on how rubber cultivation might affect biodiversity. The study by Grass et al. may be helpful in this context.

Grass, I., C. Kubitzka, V.V. Krishna, M.D. Corre, O. Mußhoff, J. Drescher, K. Rembold, P. Pütz, E. Sulpin Ariyanti, A. Barnes, U. Brose, B. Brümmer, D. Buchori, R. Daniel, K. Darras, H. Faust, J. Hein, N. Hennings, P. Hidayat, D. Hölscher, M. Jochum, A. Knohl, M. Kotowska, V. Krashevskaya, H. Kreft, C. Leuschner, N. Jun Lobite, R. Panjaitan, A. Polle, A. Potapov, M. Qaim, A. Röhl, S. Scheu, D. Schneider, A. Tjoa, T. Tscharntke & E. Veldkamp, 2020. Trade-offs between multifunctionality and profit in tropical smallholder landscapes. Nature Communications 11, 1186.

Thank you, we have now included this reference to explicitly highlight the negative biodiversity impacts of rubber (lines 293-294).

(7) L73 ‘... with palm oil and soy accounting for seven to eight times more deforestation than rubber’
To me, the wording appears not entirely clear. Does that mean palm oil and soy combined? Or each of them palm oil and soy?

Apologies, the wording was unclear. We have now amended the sentence to clarify the meaning (it is each of them – the data suggest that palm oil accounts for 8x more deforestation than rubber, and soy for 7x more).

(8) L74 I personally don’t find the nutmeg comparison very convincing

We have taken this statement out now.

(9) L75 I feel: one ‘the’ too much

Thank you. We have corrected the sentence.

(10) L621 available comes twice

Thank you. This has now been corrected.

(11) L 641 Not correctly cited. Hendaryanto is missing

Apologies. We have now corrected the reference. We have also added this reference to our main manuscript to highlight that rubber phenology can be very unpredictable (as per our response to comment 1 above).

(12) Fig 2. I don’t find the line connecting different countries appropriate. This would suggest, that there might be values somewhere in-between the countries.

We agree and have now removed the line.

Referee #4 (Remarks to the Author):

Review: New high-resolution maps show that rubber causes significant deforestation

This study proposes a new method to detect and map deforestation associated with rubber tree plantations across Southeast Asia. The authors point out that rubber crop deforestation is estimated through modeling and extrapolation, leading to high uncertainty. To overcome it, satellite observation data was processed in the cloud computing Google Earth Engine platform. Their results indicate that “deforestation for rubber is two to threefold higher than suggested by figures currently.” 3.7 million hectares of forest are suggested to be cleared for rubber plantation in Southeast Asia, according to the study’s results, 27% (i.e., 1 million hectares) in key biodiversity areas. This study deserves publication, but major issues must be improved before acceptance.

(1) First, several unclear steps in the remote sensing approach must be clarified. I provide a detailed list below, but the most critical is the lack of area adjustment using the ‘good practices’ for area estimation proposed by Olofsson et al. (2014). The whole point of this study is to reduce the uncertainty of the model-extrapolation approach, but it seems no area uncertainty from the remote sensing estimates is presented (properly).

Yes, this is a very good point. We have now added uncertainty assessments and area estimates following Olofsson et al. (2014). We now make extensive reference to this throughout the manuscript, quantifying uncertainty much more explicitly (e.g., Results lines 117-129 and 161-164, Discussion lines 226-231, caption Fig. 1, Table 1, Table 2, Methods lines 665-679, 717-725). We have also added all calculations for estimating uncertainty and sample-based area to the Supplementary Material (Extended Data Tables 1-6).

(2) Secondly, the sampling design for assessing the rubber deforestation map is also unclear.

Yes, this deserves clarification. We have now added more details on the sampling design to the manuscript (Methods lines 623-647 and 715-723). In addition, Extended Data Tables 1-6 shows the spread of reference data points across classes, and Extended Data Fig. 3 provides a map of the reference ground points.

(3) Third, it seems that uncertainty varied across the study area (mainly due to the availability of satellite observation). This must be quantified and presented because there are policy implications in areas where rubber deforestation cannot be estimated accurately.

Again, this is a very good point. We have now estimated accuracy separately for mainland Southeast Asia and insular Southeast Asia (Extended Data Tables 2 and 3) and make clear throughout the manuscript (e.g., in Results lines 119-129, Discussion lines 226-234, Methods lines 666-670, and caption Fig. 1) that uncertainty is lower for insular Southeast Asia where cloud cover and a less predictable rubber phenology may have led to omission errors. (See also point 1 in response to reviewer 3).

(4) Finally, there are missing details of the methodology used to detect and map rubber deforestation (see more information below). I could not assess the map results because the scripts and maps were not shared - I suggest the authors include them in the new version of the manuscript. See below a list of other major and minor issues that must be addressed.

Apologies. We have been trying to find a way to share the GEE repository without making the scripts public at this stage (while they are still under peer-review; it goes without saying that

everything will be public once published, and we will also make the manuscript open access and develop Earth Engine Apps to ensure easy access to the data for non-technical stakeholders). Setting up a private reviewer account proved to be challenging due to the requirement for phone or email verification by the originator of the account. However, we are now sharing the scripts as text files (uploaded to the submission portal as a zip file containing all individual scripts and as a PDF containing all scripts in a merged document).

Major and Minor Issues:

(5) Line 26: Please, define high-resolution since there is no standard for it. Landsat is called 'medium spatial resolution' (see: <https://www.frontiersin.org/articles/10.3389/frsen.2022.894571/full>), and Sentinel-2 fits this class of Earth Observation data.

We have now done this.

(6) Line 31: Please, put the rubber deforestation estimates in a regional and global context. What percent does it represent relative to the total deforestation in the study area?

We have now put our deforestation estimate in context in the caption of Table 1.

A direct comparison is challenging due to the different baselines. Taken at face value, rubber deforestation mapped in this study accounts for 5-6% of the total annual deforestation of over 3 million ha yr⁻¹ between 2001-2019 in for Southeast Asia (e.g., reported by Feng et al. (2021) *Nature Sustainability*: 4, 892-899). However, a comparison with this and other studies may underestimate rubber impacts because while we employ a baseline of 1993 the earliest baseline in most other studies is 2000. Hence Feng et al. (2021) and most other studies will include more plantation rotation than included into our deforestation estimate for post 2000.

(7) Lines 34-36: I suggest the authors be more propositive and point to domestic and global policies. The conclusion statement in the Abstract is quite general. Which ongoing supply chain deforestation-free forum could it fit? What is the implication of rubber deforestation on carbon emission estimates? Are countries neglecting it in their CO2 emission reports?

Thank you. We have now updated and strengthened the description of the policy relevance of this work in the Introduction (lines 74-91) and Discussion. The policy landscape is changing very rapidly, which is why we left the sentence in the abstract more general – but we now describe in the Introduction that a pre-print version of this manuscript formed part of the evidence contributing to policy debates over the EU Deforestation Regulation, which following prolonged debate now do include rubber. We also point to relevant deforestation-free forums such as the Global Platform for Sustainable Natural Rubber and the Forest Stewardship Council (Discussion), who have already expressed interest in the use of our data.

The implications for carbon emissions are more complex and strongly depend on what type of landcover rubber is replacing. With around 130 Mg C per ha (Petsri et al. 2013. *Journal of Cleaner Production* 52, 61-70) the carbon stored by mature rubber plantations is not insignificant and can be on a par with secondary or dry forest. In fact, in China large land carbon sinks estimated from atmospheric carbon dioxide data over China (Wang et al. 2020. *Nature* 586, 720-723) coincide with

areas where there is significant rubber expansion. Consequently, to err on the side of caution, we are not pointing out the potential neglect of rubber-related CO2 emissions in national reporting.

(8) Line 87-89: It has been said previously that rubber plantation is spectrally indistinguishable from tropical forests. What makes it possible now? Any specific Sentinel-2 spectral band can do the job?

We have now added an explanation on lines 95-96 (what makes this possible now is mainly the high resolution of Sentinel-2, which is better suited to capture smallholder plantations).

(9) Line 91-93: The statement “We track the deforestation...” shows up abruptly. How does it link with the Sentinel-2 analysis?

Thank you for pointing out the lack of clarity. We have now revised that paragraph.

(10) Line 93-96: Please, define ‘deforestation’ in the context of this study. It is unclear. This whole sentence of this last paragraph is confusing.

We have now revised this entire paragraph to increase clarity. In addition, the Supplementary Material contains a note (Extended Data Note) that discusses the definitions of ‘deforestation’ and ‘forest’ and the critical importance of clear definitions for these terms.

(11) Line 101: Please, include the user’s accuracy. Explain briefly which statistical protocol to assess the accuracy was used. Also, explain how the rubber deforestation area was estimated. Simply counting pixels lead to a biased estimate which can be adjusted using the error matrix (see: Olofsson et al., 2004).

Yes, this is a good point. We are now consistently reporting both mapped and sample-based area with associated confidence intervals. All accuracy and area estimation estimations follow Olofsson et al. (2014). This is now made clear throughout the manuscript, and we have added all details of these calculations (Extended Data Tables 1-6). (See also response to comment 1 above). Where there is a mismatch between mapped and sample-based area estimates we have added explanations for why we think this is the case (for example in the Results on lines 125-129, Discussion 228-232, Methods 668-681, caption Extended Data Table 4).

(12) Figure 1. Please replace ‘500 m resolution’ with ‘500 m pixel size’ for clarity and in ‘500 m’. The maps are blurred, lack a scale north arrow, and are too small. Include important geographic location and ocean name. The cartographic design is poor. I suggest expanding the map to fit one page.

Thank you. We have now corrected the text in the caption and improved the map in line with these suggestions.

(See also point 4 in response to reviewer 2).

(13) Line 121: Please, clarify how the NBR index was used in this part of the manuscript.

We have now provided further clarification on this (lines 142-145) and added a figure to the Supplementary Material (Extended Data Fig. 5) for clearer illustration.

(14) Figure 3. Why did you use curved lines in this graph? Did you fit a model to estimate the area? Do not smooth the statistics for aesthetic purposes. What is the uncertainty of the estimate per country? I suggest to breakdown the accuracy assessment per country since the results vary across the countries due to satellite data availability, especially in the study area.

We have now changed the figure and use bar charts instead of curved lines. Unfortunately, we do not have sufficient sample points for uncertainty estimates per country, but we have now broken down the uncertainty estimate by region (mainland Southeast Asia versus insular Southeast Asia) as per our response to comment 3 above.

(15) Line 216: What is the uncertainty of the ESA tree cover map? This might affect the overall result.

This is a good point. The ESA tree cover map achieved an overall accuracy of 74.4 (80.7 \pm 0.1 95% CI in Asia) with a reasonably good accuracy for tree cover (User's accuracy of 80.1 \pm 0.1 and Producer's accuracy of 89.9 \pm 0.1). Thus, there was a slight overestimation of tree cover. We would not expect this to affect our rubber map as it is unlikely that an area that is not tree cover would by coincidence display the spectral signature characteristic for rubber. It may affect the sample-based area corrections (in that the 'other tree cover' class would be overestimated) but this is very speculative.

We have now added a sentence on the accuracy of the ESA tree cover map to the Methods (lines 611-613), along with a reference to this independent evaluation.

(16) Line 218: Cloud cover is the most deleterious problem for assessing land cover change in Southeast Asia with optical remote sensing, including Landsat and Sentinel-2. This needs to be treated adequately in this study.

We are now treating this issue by providing separate accuracy estimates for the two subregions (mainland and insular Southeast Asia), highlighting that estimates are more uncertain in insular Southeast Asia, which is partly due to cloud cover (see response to comment 3 above and response to comment 1 reviewer 3). We also quantified the issue by country (percentage of study area per country obscured by clouds; Extended Data Table 7) and report on this in the Discussion on lines 232-234.

Regarding the methods: For sentinel-2 imagery (rubber map), we (1) used a cloud removal algorithm based on the 'QA60' cloud mask band and the 'Sentinel-2 cloud probability' datasets, and (2) derived 2-year imagery composites for rubber defoliation and refoiliation; for Landsat (deforestation map) we used (1) the Landsat CFMASK band to remove clouds/shadows and (2) annual composites. This is reported in the Methods on lines 558-567.

Line 220: What is the percent of the ground observed area?

More than 50% of the sample reference points are ground observations. We now specify this in the Methods (lines 633-636) and also added a qualifier that although the classification of these points is likely to be highly accurate, they will inevitably to some extent have an accessibility bias despite

the random sampling design. We also discuss how this may affect the validation of our rubber map (lines 674-681).

(17) Line 221: What do you mean by 'potentially' more accurate? The map is or isn't more accurate. Disaggregate your reference data used to assess the map accuracy and quantify how the map accuracy varies across the countries).

Yes. We have now quantified subregional accuracy (as above) and removed the word 'potentially'.

(18) Line 493: Please, justify the seven vegetation indices used. NDWI, NBR, and MNBR are not vegetation indices. So, why were they used? Isn't their redundancy amongst the vegetation indices used? I believe so. What is the rationale for selecting them?

We used all available bands and standard spectral indices to maximise classification accuracy but as pointed out by the reviewer, not all are vegetation indices (although some non-vegetation indices such as the NBR proved helpful) and there is inevitably an element of redundancy. To be clearer we now call them spectral indices. We also quantify their respective importance in deriving the Random Forest Classification (Extended Data Table 8) to address concerns around redundancy. The Random Forests classifier develops (uncorrelated) regression trees by picking from a random subset of candidate predictor variables (and sampling with replacement), i.e., the algorithm does tolerate a larger number of input variables, even if there is some redundancy.

(19) Method: Please, include a detailed workflow of the processes and details about the data sets used and the parameters of the main algorithms.

Thank you. We included a detailed workflow of the processes and datasets used (Extended Data Fig. 4) and added a table summarising all hyperparameter settings of the main algorithms (Extended Data Table 9).

(See also point 3 in response to reviewer 3.)

(20) Line 509: Examples of phenology spectral curves of the rubber and forest trees must be provided for representative sites.

We agree and have now added examples of spectral curves for rubber, evergreen forest and deciduous forest (Extended Data Figs. 1 and 2).

(21) Line 525-527: Apparently, only NDV was used to classify rubber and non-rubber trees. Please, clarify it and indicate how the other spectral indices were applied.

Apologies, this should have been explained more clearly (which we have now done on lines 613-620). The NDVI percentiles were used as thresholds for generating the composite images (to reduce noise and spikes in the data due remaining clouds or shadows that were not identified in the removal process). The composites used in the classifier contained all spectral indices.

(22) Line 535: Please, replace 'ground-truthed points' with 'reference ground data.'

Thank you, we have corrected this throughout.

(23) Line 537-539: Please, provide more details about the sampling design. How did you get the 1,000 points for each class?

We now include more details on the sampling design and the origin of the points (on lines 623-633). Specifically, we had a total of 3,826 reference sample points (2,010 for rubber and 1,816 for evergreen forest). The 1,000 points for each class were based on randomly sampled reference ground data collected by the World Agroforestry Centre in 2010. These covered the entire region and consisted of a mix of field data and visually interpreted very-high resolution satellite data. We updated the classification for these points for 2021 following a visual interpretation protocol (more details below). The remainder were points from more recent randomly sampled reference ground data covering mainland Southeast Asia.

(24) Line 540: Please, provide examples of images used in the visual interpretation of rubber and forest and more detail about the visual interpretation protocol. How many image interpreters? How was the reference data assessed? What are the labeling criteria?

We have now included all requested details on the visual interpretation process (Methods lines 637-647) along with example images and labelling criteria (Extended Data Fig. 6).

(25) Line 547: The high-resolution imagery from the Google Earth base map does not have the image acquisition date. How did it affect the interpretation? Justify the reference data splitting threshold.

We additionally used Google Earth Pro where we were able to see image acquisition dates. We have now included this detail (Methods lines 645-647, Extended Data Fig. 6), and we have also added a justification for the data splitting threshold (Methods lines 626-628). Importantly, the splitting threshold left us with sufficient reference data for achieving a standard error of the overall accuracy of $SE=0.01$.

(26) Line 571: The 5x5-pixel majority filter changes the border, which affects the area estimate. Please, assess the effect of this filter on the area estimate by comparing the area with and without the filtering process.

The 5 by 5-pixel majority filter might change the border, but to the best of our knowledge it would not affect the area estimates. The area of the map was calculated in Google Earth Engine based on a weighting system. For pixels on the border, the area of these pixels was only counted for the part of the pixel that was inside the border.

(27) Line 592: Please, add a figure showing examples of deforestation breakpoints, including the minimum threshold imposed. By the way, the rationale of this threshold needs to be clarified, and the figure can help explain it.

We have added Extended Data Fig. 5 to show an example of deforestation breakpoints and we now also clarify the rationale of the threshold (Methods lines 711-714). We tested a wide range of thresholds and selected the one that provided maximum overall accuracy based on the reference sample data.

(28) Line 597: What is the implication of excluding deforestation later than 2016? Does it imply that the algorithm produces maps of rubber deforestation five years out of date?

Yes, that is correct. Rubber takes around five years to mature and before maturation it is challenging to detect from earth observation. We are making the conservative assumption that our rubber map only includes plantations that are 5 years or older. Hence, we treat any detected deforestation post 2016 as a potential classification error and remove it. This does indeed mean that deforestation can only be tracked (with higher accuracy) with a delay of around five years. However, since rubber takes around 7 years to mature, any rubber entering the market now would have been planted at the latest around 2016. Hence our rubber and deforestation maps can contribute to due diligence and compliance checks for rubber that is currently traded.

(29) Data and code availability: Provide the links to access the dataset and map results and details of Google Earth Engine scripts to access the satellite and geospatial data sets used in this research.

It was impossible to assess these data sets since the links were not shared.

Apologies. Links to the dataset, map results and GEE scripts will be live in the published version of the MS. To facilitate pre-publication access, scripts are now provided as text files. Please contact us (aahrends@rbge.ac.uk) if further access is required (e.g. to our GEE repository).

Reviewer Reports on the First Revision:

Referees' comments:

Referee #1 (Remarks to the Author):

The authors have made a thorough revision and have comprehensively dealt with the small number of issues I had raised from the original version.

Referee #2 (Remarks to the Author):

The revisions made towards my comments are sufficient. I believe the paper has sufficient quality to be published in its current form.

Referee #3 (Remarks to the Author):

Dear authors,

Thank you for the revision.

I am not entirely convinced by some of the responses. In my previous review I wrote:

1) The approach is based on rubber leaf phenology. Unfortunately, very little is known or understood about rubber tree's leaf phenology. The study uses a classification into regions, where rubber defoliation occurs during January-February (region-A), and where rubber defoliation occurs in June-September (region-B). The Extended Data Fig.2 shows these rubber phenology regions.

The A and B, black or white ... approach is of course very simple, not to say simplistic. I miss a discussion and further exploration of this topic.

Even on land, there is often a suggested abrupt change from presumed leaf shedding in June-September and January-February; e.g. on the islands Sumatra, Borneo and Papua-New Guinea. Along this very long dividing line and in an unknown extent of its corridor, there could be a huge uncertainty in classification.

In the response you stated that these topics were included in the discussion (L. 599-604). L. 599-604 read:

In most of the country the lowest monthly precipitation occurs during June-September. Based on regional rubber phenology studies^{8,9} we made the necessarily simplistic assumption that in these areas the defoliation occurs during June-September with subsequent refoliation during October-December. We refer to the areas where rubber defoliation occurs during January-February as region-A, and where rubber defoliation occurs in June-September as region-B (Extended Data Fig. 3).

To me, this is more a recap of methods. I also do not concur with the assessment at country level (e.g. Malaysia and Indonesia have areas in A and B). The references 8 and 9 both come from Sumatra and data were obtained quite locally. What is known about rubber tree's leaf phenology in the study region?

The question of the very long dividing line (region A vs. region B) has not been addressed. This line runs across the islands Sumatra, Borneo and Papua-New Guinea. Along this line and in an unknown extent of its corridor, there could be a huge uncertainty in classification and rubber detection. In such a corridor: How different are predictor values from values in the training data?

Or: Is there an estimation how uncertainties change with distance to the dividing line?

In the new version of the study, two different levels of uncertainty were derived for mainland and insular Southeast Asia. What are the implications for further use of the product (map)? E.g. for country-wise comparison?

Referee #4 (Remarks to the Author):

Reply to the second review: New high-resolution maps show that rubber causes significant deforestation.

The authors answered my questions e comments carefully. The rubber mapping methodology is more precise and replicable, the discussion about the potential limitation of the results improved, and so have maps and figures. Given the relevance of this study, I recommend the manuscript for publication. If accepted, I suggest making the GEE scripts public.

Author Rebuttals to First Revision:

Referee #1 (Remarks to the Author):

The authors have made a thorough revision and have comprehensively dealt with the small number of issues I had raised from the original version.

We thank the reviewer for the helpful feedback.

Referee #2 (Remarks to the Author):

The revisions made towards my comments are sufficient. I believe the paper has sufficient quality to be published in its current form.

We are very grateful to the reviewer for the helpful feedback.

Referee #3 (Remarks to the Author):

Dear authors,

Thank you for the revision.

I am not entirely convinced by some of the responses. In my previous review I wrote:

1) The approach is based on rubber leaf phenology. Unfortunately, very little is known or understood about rubber tree's leaf phenology. The study uses a classification into regions, where rubber defoliation occurs during January-February (region-A), and where rubber defoliation occurs in June-September (region-B). The Extended Data Fig.2 shows these rubber phenology regions.

The A and B, black or white ... approach is of course very simple, not to say simplistic. I miss a discussion and further exploration of this topic.

Even on land, there is often a suggested abrupt change from presumed leaf shedding in June-September and January-February; e.g. on the islands Sumatra, Borneo and Papua-New Guinea. Along this very long dividing line and in an unknown extent of its corridor, there could be a huge uncertainty in classification.

In the response you stated that these topics were included in the discussion (L. 599-604). L. 599-604 read:

In most of the country the lowest monthly precipitation occurs during June-September. Based on regional rubber phenology studies^{8,9} we made the necessarily simplistic assumption that in these areas the defoliation occurs during June-September with subsequent refoliation during October-December. We refer to the areas where rubber defoliation occurs during January-February as region-A, and where rubber defoliation occurs in June-September as region-B (Extended Data Fig. 3).

To me, this is more a recap of methods. I also do not concur with the assessment at country level (e.g. Malaysia and Indonesia have areas in A and B). The references 8 and 9 both come from Sumatra and data were obtained quite locally. What is known about rubber tree's leaf phenology in the study region?

The question of the very long dividing line (region A vs. region B) has not been addressed. This line runs across the islands Sumatra, Borneo and Papua-New Guinea. Along this line and in an unknown extent of its corridor, there could be a huge uncertainty in classification and rubber detection. In such a corridor: How different are predictor values from values in the training data? Or: Is there an estimation how uncertainties change with distance to the dividing line?

In the new version of the study, two different levels of uncertainty were derived for mainland and insular Southeast Asia. What are the implications for further use of the product (map)? E.g. for country-wise comparison?

We thank reviewer 3 for these further helpful comments. We have made corresponding changes to the manuscript, which we outline below to clarify these points. While the link between drought and/or cold stress and leaf fall in rubber during the dry season in mainland Southeast Asia is well established, the issue of unpredictable rubber tree phenology in insular Southeast Asia is an important point and we have now added further references, quantifications and maps to expand on this issue.

Just to preface our specific response, I would like to stress that the manuscript makes clear the lower ambiguity in quantifying the distribution of rubber in climate Region-A (mainly mainland Southeast Asia) due clear-cut seasonality, compared to the greater difficulty of mapping rubber in climate Region-B (most of insular Southeast Asia) where there is lower seasonality. We have done further work to quantify this difference in accuracy (Extended Data Figs. 4-8) and added more references and text (for specific changes see below), all of which show full agreement with the points made by the reviewer and the wider sentiment we have previously expressed in the manuscript. The headline message of the manuscript is robust - namely that rubber deforestation has been substantially underestimated. In the manuscript, we make clear that our deforestation estimates are conservative.

In terms of specifics - we have added citations to four more relevant reports on rubber tree phenology in insular Southeast Asia (Methods). The available evidence all points at the more unpredictable nature of rubber phenology where there is limited seasonality. In areas (or years) without a distinct dry season, rubber phenology is reported to be highly variable, even amongst the same stand (Niu et al. 2017), and leaf shedding may be incomplete (Niu et al. 2017) or lacking altogether (Sari et al. 2021).

Where a phenology signal is reported (Alchemi et al. 2022, Azizan et al. 2023, Azizan et al. 2021, Niu et al. 2017, Razak et al. 2017), the reports support the time windows we assume for Region-A and Region-B. However, we agree that the available literature contains a spatial study bias; reports for insular Southeast Asia cover only around 18 sites and are focussed on the main rubber growing areas (Sumatra, Malay Peninsula), with much fewer reports for Boreno and islands further east – a challenge that we now specifically articulate in the Methods (see below).

Niu, F., Röhl, A., Meijide, A., Hendrayanto & Hölscher, D. Rubber tree transpiration in the lowlands of Sumatra. *Ecology* 10, doi:10.1002/eco.1882 (2017).

Alchemi, P. J. K. & Jamin, S. Impact Of Pestalotiopsis Leaf Fall Disease On Leaf Area Index and Rubber Plant Production. *IOP Conference Series: Earth and Environmental Science* 995, doi:10.1088/1755-1315/995/1/012030 (2022).

Azizan, F. A., Astuti, I. S., Young, A. & Abdul Aziz, A. Rubber leaf fall phenomenon linked to increased temperature. *Agriculture, Ecosystems & Environment* 352, doi:10.1016/j.agee.2023.108531 (2023).

Azizan, F. A. et al. Using Multi-Temporal Satellite Data to Analyse Phenological Responses of Rubber (*Hevea brasiliensis*) to Climatic Variations in South Sumatra, Indonesia. *Remote Sensing* 13, doi:10.3390/rs13152932 (2021).

Razak, J. A. b. A., Shariff, A. R. b. M., Ahmad, N. b. & Ibrahim Sameen, M. Mapping rubber trees based on phenological analysis of Landsat time series data-sets. *Geocarto International*, 1-24, doi:10.1080/10106049.2017.1289559 (2017).

The unpredictable nature of rubber phenology in insular Southeast Asia is closely associated with limited seasonality. Therefore, we have now added a map to the Supplementary Material to highlight the areas where there is indeed a pronounced dry season during the assumed defoliation windows in Region-A and Region-B, and, importantly, the areas where there is not – which precludes a strong phenology signal (Extended Data Fig. 4). We also added a map showing the driest month for each area (Extended Data Fig. 5). Both maps are based on 15-year monthly rainfall averages.

Combined, these figures serve to highlight that while particularly east of Sulawesi there is a lot of local variation in the onset of the dry season, which is difficult to capture in a climatic zonation, this is not the key challenge. The key challenge is the absence of a distinct dry season altogether. Areas such as Malaysian Borneo (which as the reviewer pointed out could have also been placed into Region-B) have so little seasonality that they are hard to allocate into any climatic zone. The same is true for areas such as the Makalu islands, where the monsoonal rainfall patterns are so modified that they fall outwith the pattern assumed in Region-A and Region-B, but (1) we map little rubber in these areas (Indonesia's main rubber growing area is Sumatra), and (2) the zonation becomes arbitrary in areas that are close to perhumid. Close to the equator the dry season is neither prolonged nor distinctive, as illustrated by the findings of Niu et al. (2017) who reported that soil moisture never dropped below critical levels at any of their 10 study sites on Sumatra.

Where there is no or little seasonality, our ability to map rubber is lowest, and this is also where we are most likely to underestimate deforestation. As requested, we have added more information to the Discussion on the implications of this for the use of our maps, especially regarding country comparisons (see specific text changes below).

To accommodate the question raised by the reviewer of estimating how uncertainty changes with distance from the dividing line between Region-A and Region-B we have added Extended Data Figs. 7 and 8. Specifically, Extended Data Fig. 7 shows a map of our classification errors (false positives and false negatives) and allows the reader to see how these are related to Region-A and B. Extended Data Fig. 8 shows a histogram of errors against latitude. As above, these figures highlight that the density of false negatives is highest in the areas of lower seasonality (near the equator). The figures also show that we overwhelmingly err on the side of omission errors and that the risk of commission errors is much lower.

We are grateful to reviewer 3 for these comments, as these additional analyses and additional clarification on the limitations of rubber detection in Region-B further support the important headline message of the manuscript – that rubber deforestation has been underestimated by at least two to threefold, and that the scale of this underestimation may be even higher due to the challenges of aseasonality in insular Southeast Asia.

====

Summary of substantial changes:

We added Extended Data Figs. S4-S8 to the Supplementary Material.

We added 6 further references to the Methods, four of which describe rubber tree phenology in insular Southeast Asia, with the rest discussing the evolution of the deciduous behaviour in *Hevea brasiliensis* and the environmental triggers for leaf shedding.

We have made the following text changes to the manuscript:

Introduction (lines 100-101)

We replaced “Our approach is based on the distinctive phenological signature of rubber plantations which allows them to be distinguished from both evergreen (Extended Data Fig. 1) and deciduous (Extended Data Fig. 2) tropical forests based on leaf-fall and regrowth, *which occur in specific time windows that differ by region (Extended Data Fig. 3).*”

with

[...], **which particularly in mainland Southeast Asia occur in specific time windows.**

Results

Lines 123-128

We added (text in bold):

The rubber maps achieved a good overall classification accuracy (OA=0.95 ±0.02 95% CI; Extended Data Table 1) with good accuracy and precision of estimates for mainland Southeast Asia (OA>0.99 ±0.01 95% CI; Extended Data Table 2) but higher omission errors and less overall accuracy for insular Southeast Asia (OA=0.85 ±0.06 95% CI; Extended Data Table 3). **Here, limited seasonality (Extended Data Fig. 4) and greater heterogeneity in climatic conditions (Extended Data Fig. 5) mean that rubber phenology is less predictable, with trees defoliating at different times, exhibiting partial defoliation or no defoliation at all³⁴. Hence, despite running the rubber detection algorithm separately for two different subregions to address the spatial heterogeneity in climate conditions (Extended Data Fig. 6), omission errors remain in insular Southeast Asia (Extended Data Figs. 7 and 8; Methods).**

We also added lines 131-132

The low Producer’s accuracy when based on estimated area **is in part due to us erring on the side of omission errors (mainly affecting insular Southeast Asia),**

Discussion (lines 232-243)

Second, ground reference data indicate that we err on the side of omission errors, with sample-based area estimates³³ suggesting that the rubber area could be significantly larger (Extended Data Table 1), particularly in insular Southeast Asia. **This is because the limited seasonality of the equatorial climate precludes a strong and predictable phenological response of rubber in insular Southeast Asia³⁴.** Furthermore, insular Southeast Asia has more persistent cloud cover compared with mainland Southeast Asia, with 7% and 10% of the study area in Indonesia and Malaysia, respectively, lacking clear Sentinel-2 images (Extended Data Table 7). Consequently, our maps are more accurate for mainland Southeast Asia than for insular Southeast Asia (Extended Data Tables 2 and 3), where rubber area (and hence associated deforestation) may be underestimated. **Any comparisons by country or other spatial units across these two subregions thus need to be done with caution in the light of this limitation.**

Methods

We added lines 603-607:

[...] rubber plantations shed their leaves during the dry season and subsequently regain their leaves before the onset of the wet season. **Whether this is primarily a response to drought or cold stress is**

subject of ongoing research^{7,8}, but particularly in mainland Southeast Asia the cold and dry season coincide, meaning that, here, the lack of mechanistic understanding of this phenological response does not preclude identifying the time window of its occurrence.

We provided some additional details underpinning our approach of two different regions for the analyses, and the uncertainty associated with aseasonality near the equator (lines 608-652):

While mainland Southeast Asia is characterised by a seasonal monsoonal climate, insular Southeast Asia is less seasonal and the onset of a dry season, if present, mostly falls into a different time of year compared to mainland Southeast Asia (Extended Data Fig. 5). Therefore, we divided the region into two subregions (Extended Data Fig. 6). In mainland Southeast Asia the northeast monsoon brings dry and cool continental air⁹ and rubber defoliation generally occurs during January-February with subsequent refoiliation during March-April (Extended Data Fig. 1). This distinct signature also allows the separation of rubber from deciduous forest, which is present in much of mainland Southeast Asia: leaf regrowth in other species in deciduous forest mainly coincides with the onset of the wet season in May (Extended Data Fig. 2).

In contrast to mainland Southeast Asia, large parts of insular Southeast Asia do receive rainfall during the northeast monsoon with the south-westerly flowing airmasses gathering moisture as they pass over the warm sea. Instead, there can be a dry season during the southwest monsoon (May to September) when the airmasses reverse and the north-easterly blowing winds bring dry air from the Australian continent⁹. However, in the equatorial maritime climate the dry season tends to be neither prolonged nor distinctive (Extended Data Fig. 4), and soil moisture can remain stable or at least above critical levels¹⁰.

Originating from the Brazilian Amazon, the deciduous behaviour of *Hevea brasiliensis* is thought to have evolved as an adaptive strategy for drought or more generally stress avoidance⁸. Consequently, in years or areas where there is no clear-cut stress in the form of a distinctive dry and/or cold season leaf shedding will only be partial, not take place at all, and/or will be influenced by micrometeorological conditions with trees defoliating asynchronously even within the same stand¹⁰. Few reports exist on rubber phenology in insular Southeast Asia. The limited available evidence¹⁰⁻¹⁵ (covering a total of c. 18 sites, which are spatially biased towards the main rubber growing areas Sumatra and Malay Peninsula, with only one report for Borneo and none for islands further east) suggests that where there is a predictable defoliation window, it generally occurs during January to February (Malay Peninsula and northern Sumatra) or during June to September (further south).

As the divergent defoliation patterns described in the available literature mainly affect Indonesia, and as due to consistently high temperatures stress is mainly likely to occur in the form of drought there, we delineated two climatic subzones as follows: we mapped average monthly precipitation¹⁶ across Indonesia and identified the driest month for each pixel (c. 1×1 km); we then delineated all pixels with the driest month between June and September as a separate subregion (Region-B, where defoliation was assumed to take place June to September with subsequent refoiliation during October to December). The remaining pixels and all of Malaysia and mainland Southeast Asia were assigned to Region-A, where defoliation was assumed to take place between January and February with subsequent refoiliation during March to April (Extended Data Fig. 6).

The lack of distinctive seasonality near the equator means that inaccuracy of our classification was greatest near the equator (Extended Data Figs. 7 and 8) and mainly manifested in omission errors (3% of our 661 ground reference points used for validation were false negatives and only 0.3% were false positives; of the false negatives 95% occurred in insular Southeast Asia). Beyond c. 7° north the climate becomes more continental with clear-cut seasonality and no more false negatives were recorded.

The unique phenology of rubber, **where exhibited**, thus make rubber distinguishable [.....]

For completeness we include here an equivalent of Extended Data Fig. 8 with the x axis showing distance from the divide between Region-A and Region-B instead of latitude. We have not added this figure to the manuscript as it is just another way of displaying the data underlying Extended Data Fig. 7 and it transports the same message as Extended Data Figs. 7 and 8; however, it was useful for us to produce this figure as an internal check.

Frequency distribution of omission and commission errors with increasing distance from Region-B.

A: Of n=661 validation ground reference points, 21 had a classification error, of which 90% were omission errors (false negatives) with only two commission errors (false positives). **B:** The density of omission errors was highest inside Region-B. False negatives remained up until c. 7° north (c. 500 km distance northwards from the dividing line between Region-B and Region-A). Beyond this point the climate becomes more continental and seasonal (Extended Data Fig. 4) and no more false negatives were found (Extended Data Figs. 7 and 8).

Thus, the issue is less related to the boundary (uncertainty did not increase near or at the divide between Region-A and Region-B); instead, omissions were present throughout Region-B and further north – up until the point where there is pronounced seasonal dry and/or cold stress and rubber phenology is more clear-cut.

Referee #4 (Remarks to the Author):

Reply to the second review: New high-resolution maps show that rubber causes significant deforestation.

The authors answered my questions e comments carefully. The rubber mapping methodology is more precise and replicable, the discussion about the potential limitation of the results improved, and so have maps and figures. Given the relevance of this study, I recommend the manuscript for publication. If accepted, I suggest making the GEE scripts public.

Once again, we thank the reviewer for their helpful feedback. We had in the previous submission shared the scripts with the reviewer and as previously mentioned, the scripts are uploaded to GitHub and will be publicly released once the article is accepted for publication.

Reviewer Reports on the Second Revision:

Referees' comments:

Referee #3 (Remarks to the Author):

Dear authors,

Thank you for this highly relevant study. I find the results very important. I would also hope that it triggers further studies into sustainable land management.

Best, reviewer 3

Author Rebuttals to Second Revision:

Referee #3 (Remarks to the Author):

Dear authors,

Thank you for this highly relevant study. I find the results very important. I would also hope that it triggers further studies into sustainable land management.

Best, reviewer 3

We thank reviewer 3 for their helpful feedback on the manuscript.